# Modeling the impact of drug interactions on therapeutic selectivity

Zohar B. Weinstein [1], Nurdan Kuru[2], Szilvia Kiriakov [3,4], Adam C. Palmer [5], Ahmad S. Khalil [4,6,7], Paul A. Clemons[8], Muhammad H. Zaman[7,9], Frederick P. Roth[10,11,12] & Murat Cokol[2,5]

Combination therapies that produce synergistic growth inhibition are widely sought after for the pharmacotherapy of many pathological conditions. Therapeutic selectivity, however, depends on the difference between potency on disease-causing cells and potency on non-target cell types that cause toxic side effects. Here, we examine a model system of anti-microbial compound combinations applied to two highly diverged yeast species. We find that even though the drug interactions correlate between the two species, cell-type-specific differences in drug interactions are common and can dramatically alter the selectivity of compounds when applied in combination vs. single-drug activity—enhancing, diminishing, or inverting therapeutic windows. This study identifies drug combinations with enhanced cell-type-selectivity with a range of interaction types, which we experimentally validate using multiplexed drug-interaction assays for heterogeneous cell cultures. This analysis presents a model framework for evaluating drug combinations with increased efficacy and selectivity against pathogens or tumors.

[1] Department of Pharmacology and Experimental Therapeutics, Boston University School of Medicine, Boston, MA 02115, USA. [2] Faculty of Engineering and Natural Sciences, Sabanci University, 34956 Istanbul, Turkey. [3] Program in Molecular Biology, Cell Biology & Biochemistry, Boston University, Boston, MA 02215, USA. [4] Biological Design Center, Boston University, Boston, MA 02215, USA. [5] Laboratory of Systems Pharmacology, Harvard Medical School, Boston, MA 02115, USA. [6] Wyss Institute for Biologically Inspired Engineering, Harvard University, Boston, MA 02115, USA. [7] Department of Biomedical Engineering, Boston University, Boston, MA 02215, USA. [8] Chemical Biology & Therapeutics Science Program, Broad Institute of Harvard and MIT, Cambridge, MA 02142, USA. [9] Howard Hughes Medical Institute, Boston University, Boston, MA 02215, USA. [10] Donnelly Centre and Departments of Molecular Genetics and Computer Science, University of Toronto, Toronto, ON M5S 3E1, Canada. [11] Lunenfeld-Tanenbaum Research Institute, Toronto, ON M5S 3E1, Canada. [12] Canadian Institute for Advanced Research, Toronto, ON M5S 3E1, Canada. Correspondence and requests for materials should be addressed to M.C. (email: mcokol@axcellahealth.com)

Combination therapies are common in the treatment of cancer and infectious diseases[1,2]. The use of drug combinations is motivated by evidence that they can achieve cure rates superior to monotherapies[3–5]. Drug combinations may be classified as synergistic or antagonistic when the observed effect of the combination is greater or lesser, respectively, than is expected based on the components' effects as single agents[6,7]. Consequently, much effort has been applied to the task of identifying synergistic combinations. Drug-interaction screens identify combinations with increased efficacy against specific cell lines or phenotypes[8], and computational methods aim to predict drug synergies using chemogenomics[9], genetic interactions[10], and physicochemical properties[11,12].

However, consider a drug combination that is synergistic against pathogenic or cancerous cells, but also has synergistic toxicity to healthy host cells. In this case, there will be no benefit to the therapeutic window, being the difference between the dose required for the desired effect and the dose-limiting toxicity. It is apparent that the efficacy of a synergistic combination is entirely dependent on avoiding synergistic toxicity to unintended cell types, as shown by Lehar et al.[13] and reviewed in Bulusu et al.[14]. Extending this idea, it has been debated that it is not synergy itself that is pharmacologically useful, but differential drug interactions between cell types, with the essential goal being a more favorable interaction on the target cell type than on non-target cell types[15–17].

To test this idea, we implemented an experimentally tractable system to systematically characterize how cell-type-specific drug interactions affect the selectivity of combination therapies, by profiling combinations of antifungal drugs applied to the yeast species *Saccharomyces cerevisiae* and *Candida albicans*. Individual differences in single-drug sensitivity constitute therapeutic windows that select for single cell-type; however combinations of drugs may have selectivity that varies from individual agents. We used a sensitive screen to assess all 66 pairwise interactions of 12 antifungal small molecules (henceforth "drugs") in *C. albicans*, selected for direct comparison to a recent drug-interaction data set in *S. cerevisiae*[10] to determine differences in selectivity due to single agents vs. combinations. Our model framework and subsequent mixed culture assays show that therapeutic windows may be enhanced or diminished by differential drug interactions.

## Results

**Precise assessment of drug interactions in two model yeasts**. We used a sensitive $8 \times 8$ checkerboard assay to assess all 66 pairwise interactions of 12 antifungal drugs (Table 1), in *C. albicans* and *S. cerevisiae*[10]. This screen included drugs that target DNA, cell wall, and metabolism as well as microtubule, phosphatase, and kinase inhibitors. All *C. albicans* experiments were conducted in this study. Eight *S. cerevisiae* experiments were newly conducted for this study: methyl methanesulfonate tested against itself, bromopyruvate, calyculin A, dyclonine, fenpropimorph, haloperidol, rapamycin, and tunicamycin. Other experimental data involving *S. cerevisiae* were obtained from Cokol et al., 2011. Drug interactions were quantified by isobologram analysis; briefly, each interaction score ($\alpha$) for a drug pair was calculated from the concavity of the isophenotypic contours that map regions of similar growth inhibition across the drug-concentration matrix.

To produce a reference that is by definition non-interacting, we measured "self–self" interactions (drugs combined with themselves) for ten drugs in both yeast species. This produced interaction scores tightly distributed around zero (mean = −0.01, std. dev. = 0.4) and defined 95% confidence intervals for deviation from additivity. Synergy and antagonism were thereby

identified from these confidence intervals as $\alpha < -0.8$ or $\alpha > +0.8$, respectively (Fig. 1a). Among the 66 drug combinations tested in *C. albicans*, 20 synergistic and 27 antagonistic drug pairs were identified (Fig. 1b).

Drug interactions were substantially, but not perfectly conserved between *C. albicans* and *S. cerevisiae* (Spearman correlation test $r = 0.42$, $p$-value = $1.8 \times 10^{-4}$) (Fig. 2). Synergistic, but not antagonistic, interactions significantly overlapped in these related species (Fisher's exact test, shared synergy: $p < 3 \times 10^{-5}$, shared antagonism: $p = 0.44$). Notably, nine combinations had highly divergent interactions, being synergistic in one species and antagonistic in the other, suggesting that drug combinations may be used to selectively inhibit a particular cell type.

**Cell-type selectivity of individual drugs**. In order to understand the relationship between drug interactions and cell-selective inhibition, we first considered the selectivity of individual drugs. The concentrations required for 50% growth inhibition of *C. albicans* ($IC50_{alb}$) and *S. cerevisiae* ($IC50_{cer}$) were correlated between species ($r = 0.91$, $p < 10^{-13}$; Supplementary Fig. 1), but most drugs had a therapeutic window for cell-selective inhibition due to a two-fold or greater difference in IC50 between cell types. We defined the selectivity score of a single-drug "A" ($selectivity_A$) as $\log_2(IC50_{A,alb} / IC50_{A,cer})$, such that a score of 0 indicates no selectivity, and a score of 1 (or −1) denotes that twice (or half) as much drug is required to inhibit *C. albicans* compared to *S. cerevisiae*. Neither species was on average more drug-sensitive or resistant than the other (no significant bias in selectivity scores by sign test, $p = 0.77$). Benomyl and tunicamycin had the greatest single-agent selectivity for *C. albicans* (3.8 and 2.4), while staurosporine and bromopyruvate were most selective for the growth of *S. cerevisiae* (−1.7 and −1.6).

**Drug interactions alter the selectivity of drug combinations**. We explored the impact of drug interactions on selectivity by superimposing the isophenotypic contours from drug-interaction experiments for each cell type, for the greatest level of inhibition present in interaction data sets for both species (mean inhibitory level = 0.31; std. dev. = 0.17). This visualization shows the regions of selectivity between the drug-interaction contours and allows the comparison of the selectivity of a combination with the

**Table 1 Antifungal drugs used in the study**

| Drug | Abbreviation | PubChem ID | alb IC50 µg/ml | cer IC50 µg/ml |
|---|---|---|---|---|
| Benomyl | BEN | 28780 | 130 | 9.6 |
| Bromopyruvate | BRO | 70684 | 367 | 1086 |
| Calyculin A | CAL | 5311365 | 2.1 | 2.2 |
| Dyclonine | DYC | 3180 | 5.4 | 8.9 |
| Fenpropimorph | FEN | 93365 | 0.1 | 0.3 |
| Haloperidol | HAL | 3559 | 150 | 39 |
| Methyl methanesulfonate | MMS | 4156 | 128 | 65 |
| Pentamidine | PEN | 4735 | 27 | 73 |
| Rapamycin | RAP | 5284616 | 0.15 | 0.04 |
| Staurosporine | STA | 44259 | 0.2 | 0.6 |
| Terbinafine | TER | 1549008 | 1.3 | 2.1 |
| Tunicamycin | TUN | 6433557 | 1.7 | 0.3 |

All drug names are provided as well as abbreviations used in figures, PubChem ID and IC50 (µg/mL) for each drug tested in *C. albicans* (alb) and *S. cerevisiae* (cer). IC50 levels were determined with yeast cells grown overnight and diluted in liquid culture to $OD_{600} = 0.1$. *S. cerevisiae* and *C. albicans* concentration-response experiments were conducted in parallel to allow for direct comparison of IC50 levels

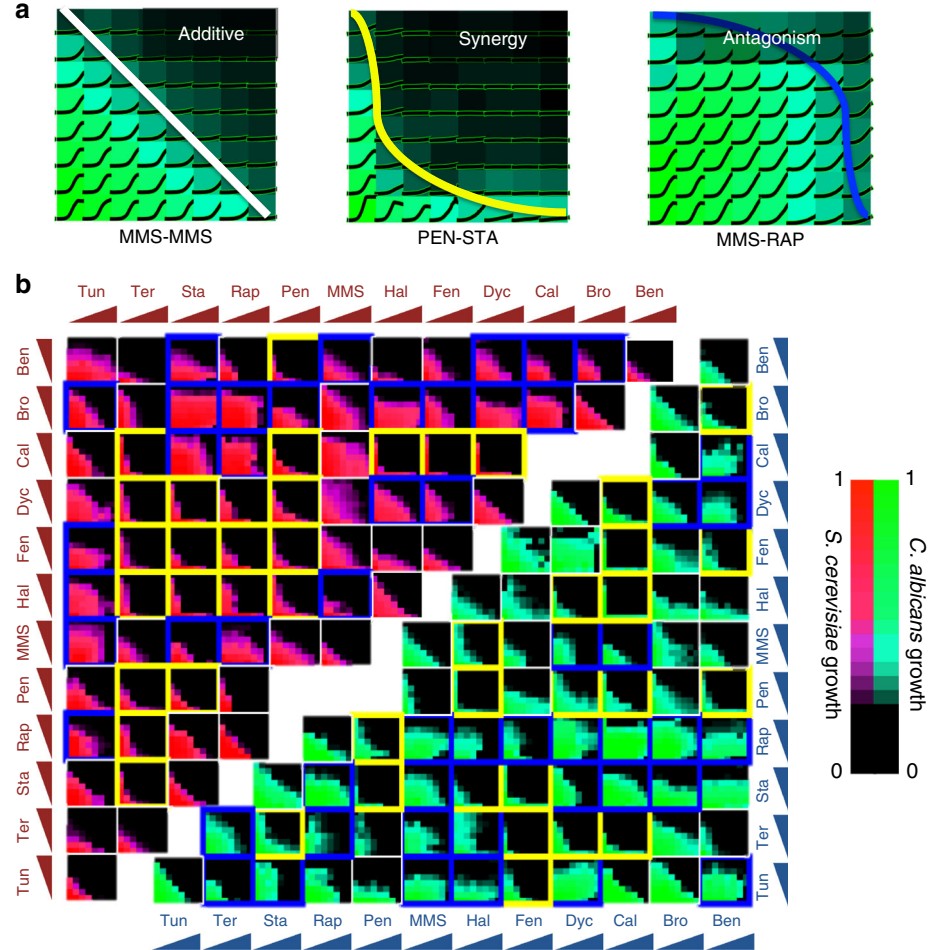

**Fig. 1** Drug-interaction screen results in *C. albicans* and *S. cerevisiae*. **a** Classification of drug interactions. Yeast cells were grown in a 2D grid with linearly increasing concentrations of one drug on each axis. Representative experiments on *C. albicans* with growth curves are shown on a heat map for growth. Drug interactions are assessed based on the concavity of isophenotypic contours. On the left, a "self–self" experiment is shown, wherein MMS is increased on each axis. Such pairs are defined as additive and have linear isophenotypic contours. Drug pairs with concave (PEN-STA) or convex (MMS-RAP) contours are defined as synergistic or antagonistic, respectively. **b** Drug-interaction experiments among all pairwise combinations of 12 drugs in *C. albicans* and *S. cerevisiae*. Each subplot shows the growth level for each drug-concentration combination for *C. albicans* (green/teal) and *S. cerevisiae* (red/magenta). Synergistic or antagonistic drug pairs are shown in yellow or blue boxes, respectively. Eight *S. cerevisiae* experiments were newly conducted for this study: MMS tested against MMS, BRO, CAL, DYC, FEN, HAL, RAP, and TUN. Other experimental data involving *S. cerevisiae* were obtained from Cokol et al., 2011

selectivity of individual drugs (Fig. 3, left panel). Selectivity of a combination "A + B" was determined similarly to the single-drug selectivity, by the ratio of total drug concentrations required to achieve an equal level of inhibition (Methods), where selectivity$_{A+B} = \log_2(IC_{A+B,alb}/IC_{A+B,cer})$, similar to a previously defined selectivity index[13] (Supplementary Fig. 2, upper panel). Selectivity of self–self combinations was almost perfectly correlated with their single-drug selectivity ($r = 0.99$, $p = 3.6 \times 10^{-5}$), as expected from first principles.

Combination selectivity was unaltered when two drugs have self–self or additive interactions (e.g., methyl methanesulfonate–methyl methanesulfonate, Fig. 3a, left). However, for drug pairs whose interactions vary between cell types, the selectivity of the combination diverged from what is anticipated from its component drugs. Pentamidine and staurosporine (PEN + STA) each preferentially inhibit *C. albicans*, and because they are synergistic only in *C. albicans*, their combination enhances selectivity for the growth of *S. cerevisiae* compared to either single drug (Fig. 3b, left). Antagonistic interactions can also enhance selectivity: methyl methanesulfonate and rapamycin (MMS + RAP) each preferentially inhibit *S. cerevisiae*, and in

combination produce an especially strong antagonism in *C. albicans* which enhances their selectivity for the growth of *C. albicans* (Fig. 3c, left). Differential interactions can both strengthen and weaken selectivity: pentamidine and fenpropimorph (PEN + FEN) both preferentially inhibit *C. albicans*, but are antagonistic only in this cell type (Fig. 3d, left), which causes diminished selectivity. A yet more striking result ensues from the divergent interactions of calyculin A and dyclonine (CAL + DYC): though each drug alone preferentially inhibits *C. albicans*, their combination demonstrates such potent synergy only in *S. cerevisiae* that their cell-type selectivity is inverted and is therefore expected to select for the growth of *C. albicans* (Fig. 3e, left).

In order to compare the observed selectivity of combinations with a null model, we approximated expected selectivity (selectivity$_{exp}$) as the combination selectivity that would be observed if drugs A and B have additive interactions in both species (Fig. 3, middle column) (Supplementary Fig. 2, lower panel). Comparing expected and observed selectivity across the complete set of drug combinations (example comparisons in Fig. 3, right column), we found that 41 of 66 combinations showed a significant difference in selectivity from additive

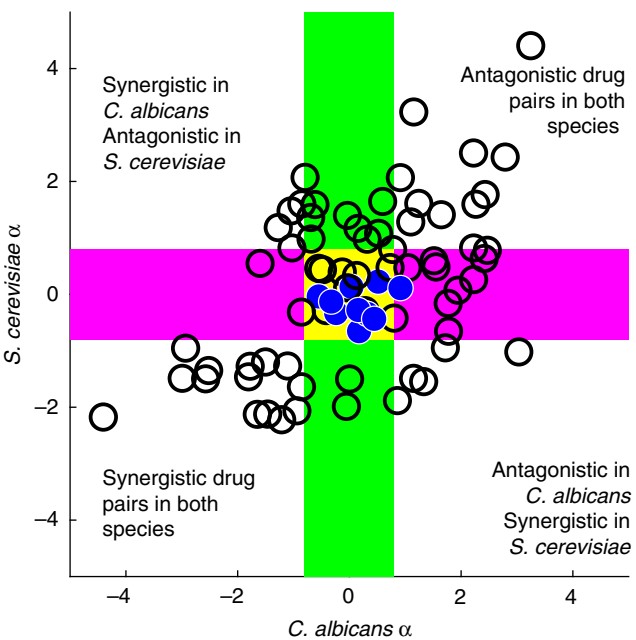

**Fig. 2** Conserved and divergent drug interactions between cell types. A comparison of α scores in two yeast species. Based on the 95% confidence interval of self–self interactions, interaction scores are interpreted as follows: $α < −0.8$ as synergistic, $α > +0.8$ as antagonistic. Additive drug combinations are within the magenta shaded area on the y-axis for *S. cerevisiae* and within the green shaded area on the x-axis for *C. albicans*. Self–self drug pairs are marked in blue and are centered on zero for both axes, as expected. α scores in *C. albicans* and *S. cerevisiae* significantly correlate, showing that drug interactions are conserved between these two distantly related yeast species (Spearman correlation test $r = 0.42$, $p$-value $= 1.8 \times 10^{-4}$). However, not all drug pairs followed this trend. At the upper left and lower right quadrants are highly divergent drug pairs, showing positive interactions in one species and negative interactions in the other

expectation based on self–self experimental variance (mean = 0.12, std. dev. = 0.10) (Supplementary Fig. 3). Thus, differential drug interactions powerfully and quite commonly influence the cell-type specificity of drug combinations, with the effect of enhancing or diminishing therapeutic windows.

A common goal of drug combination design is to identify drug pairs that synergistically inhibit the intended target cell type while not producing synergy in other cell types. However, in our study we observed that drug pairs with selectivity for the growth of *S. cerevisiae* or *C. albicans* may be synergistic, additive or antagonistic in either species (Supplementary Fig. 4). Accordingly, drug combinations showing significant selectivity against one species were not enriched for synergistic interactions in that species (Fisher's exact test, $p > 0.05$). We hypothesized that selectivity is associated with the difference of drug interactions between cell types. As a measure of drug-interaction difference, we calculated delta-α, the difference of α scores between *C. albicans* and *S. cerevisiae* ($α_{alb} − α_{cer}$). Delta-α scores are high for pairs that are antagonistic against *C. albicans* and synergistic for *S. cerevisiae*. Overall, there was a weak but significant correlation (Spearman correlation test, $r = 0.26$, $p = 0.02$) between cell-selective growth inhibition and delta-α (Supplementary Fig. 5). Therefore, we conclude that combinatorial selectivity is influenced by the difference of two-drug interactions, neither of which is necessarily synergistic.

In order to understand the effect of antimicrobial resistance on therapeutic selectivity, we modeled the effects of 100-fold resistance on selectivity metrics for all tested drug pairs.

We assumed that isophenotypic contours scaled with changes in drug sensitivity[18] and simulated resistance by multiplying the minimal inhibitory concentration of one compound by 100 while preserving the shape of the drug-interaction isobole. We observed that delta-α and sel–sel$_{exp}$ are not significantly correlated after simulating for resistance, suggesting that extreme drug resistance is more influential on selectivity than variation in drug interactions (Supplementary Fig. 5).

**Validation of the selectivity model in co-cultures**. Here we have modeled the selectivity of combinations of drugs to different fungal species. However, it is worth noting that sensitivity to drug combinations was tested separately for each species, and not together. To experimentally test the predicted selectivity change due to drug interactions, we conducted co-culture assays with fluorescently labeled strains of *S. cerevisiae* (mCherry + , GFP−) and *C. albicans* (GFP + , mCherry−). We created a mixed culture of two fluorescently labeled yeast species with approximately equal number of cells from both species based on flow cytometry. Mixed cultures were treated with two individual drugs or their combination, incubated for 4 h, and assessed with flow cytometry for the %*C. albicans* and %*S. cerevisiae* after treatment (Fig. 4a). Drug-free controls were used as a reference to confirm single-drug selectivity in the context of yeasts with different growth rates. For each experiment, we computed a selectivity score following the same formula as our model: log$_2$(*C. albicans*/ *S. cerevisiae*). Importantly, since the growth rate of *C. albicans* is faster than *S. cerevisiae*, it is expected that the %*C. albicans* in the no drug condition will increase as compared to the initial ratio.

In these experiments, we used two-drug pairs with striking phenotypes illustrated in Fig. 3: (i) CAL + DYC is synergistic in both species but the synergy is stronger in *S. cerevisiae*. According to our model, each of these drugs is expected to select for *S. cerevisiae*, however the combination is expected to select for *C. albicans* due to the inverted selectivity (Fig. 3e). Figure 4b confirms the expectation that %*C. albicans* in the no drug condition increases in the absence of selective pressure. In agreement with the single-species experiments, the selectivity scores for the CAL or DYC treated cultures were lower than the no drug condition, indicating that each of these drugs selects for *S. cerevisiae* growth. The combination CAL + DYC had a higher selectivity score than the no drug condition, indicating that the combination selects for *C. albicans*, thereby validating the prediction of inverted selectivity under treatment with the CAL + DYC combination. (ii) MMS + RAP is antagonistic in both species but the antagonism is stronger in *C. albicans*. According to our model, each of these drugs is expected to select for *C. albicans*, and the combination is expected to have a higher selectivity for *C. albicans* than either drug, due to enhanced selectivity by antagonism (Fig. 3c). In agreement with our model, we observed that the selectivity score for MMS or RAP were higher than the no drug condition, indicating that these drugs individually select for *C. albicans*. The selectivity score for the combination MMS + RAP was higher than either single drug, validating the predicted enhanced selectivity by antagonism (Fig. 4c). All data collected for these experiments are presented as Supplementary Fig. 6.

In order to extend our approach to alternative phenotypes, we conducted drug-interaction experiments for fungicidality in mixed cultures of fluorescent *C. albicans* and *S. cerevisiae* (Fig. 5a). Among all 12 drugs studied, only MMS and RAP exhibited acute strong fungicidal activity, hence, were amenable to an assay of selective cell killing (Supplementary Fig. 7). This combination is antagonistic in both *C. albicans* and *S. cerevisiae*, but our analysis suggested enhanced selectivity for the growth

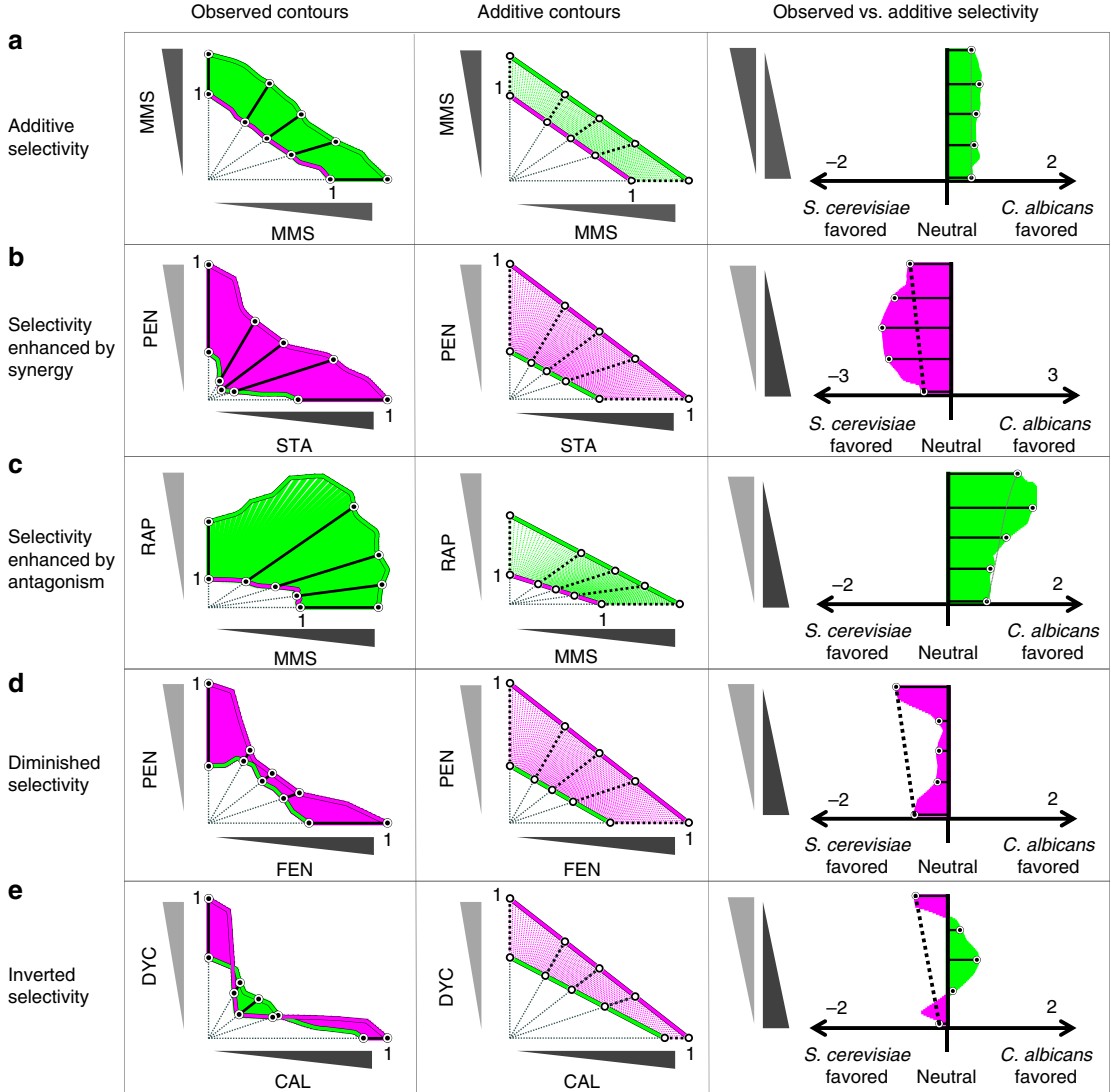

**Fig. 3** Drug interactions may enhance, diminish or invert selectivity. Left panel: Observed isophenotypic contours of drug-interaction assays for *S. cerevisiae* (magenta) and *C. albicans* (green) are overlaid in a 2D grid adjusted for relative concentration. We linearly transformed the isophenotypic contours for drug-interaction assays so that *S. cerevisiae*'s isophenotypic contour intercepted both *x* and *y* axes at 1. Selectivity of a combination was determined by the log ratio of the distance from the origin to the *C. albicans* vs. *S. cerevisiae* contours ($\log_2(d_{albicans}/d_{cerevisiae})$). Selectivity is therefore positive for drugs or combinations that select for *C. albicans*. Middle panel: the middle panel demonstrates the null model for expected combination selectivity, assuming drug pairs are additive. As in observed selectivity, expected selectivity is calculated based on the log ratio of distances from the origin with positive expected selectivity corresponding to an expected selectivity for *C. albicans*, in the absence of synergistic or antagonistic drug interactions. Right panel: observed (solid) and expected selectivity (dashed line) scores are overlaid. Green- or magenta- shaded regions represent observed *C. albicans* or *S. cerevisiae* selectivity. Deviations from expected selectivity indicates a change in selectivity due to drug interactions. Five representative drug pairs are shown with expected (additive) selectivity (**a** MMS + MMS), enhanced selectivity (**b** pentamidine and staurosporine and **c** MMS and rapamycin), diminished selectivity (**d** fenpropimorph and pentamidine), and inverted selectivity (**e** calyculin A and dyclonine), compared to single-agent expected selectivity

of *C. albicans* due to a difference in the strength of antagonism (Supplementary Fig. 8). We assessed drug interactions for fungicidal activity by co-culturing yeast strains in a 5 × 5 combination matrix of MMS and RAP for 1 h, plating cells and enumerating cell killing by counting fluorescent colony-forming units (CFU).

In strong agreement with the single-species drug-interaction experiments (Fig. 3c), we observed that MMS + RAP is antagonistic for fungicidal activity in both species, but to a stronger degree in *C. albicans* (Fig. 5b, c, Supplementary Fig. 9). Consistent with the superimposed growth isoboles, we observed that the low MMS-high RAP region is powerfully selective, killing more than 99% of *S. cerevisiae* cells with less than 50% fungicidal

effect on *C. albicans*. Importantly, each of MMS and RAP alone have similar fungicidal concentrations for both species, and are incapable of exerting such effective cell-selective killing as single agents.

## Discussion

The ultimate goal of synergistic drug combinations is to enhance the therapeutic window between efficacy and toxicity[13]. Drug-interaction screens may identify combinations with increased efficacy and selectivity for specific cell lines or phenotypes[8]. In this study, we showed that while synergistic combinations can indeed increase the cell-type selectivity of growth-inhibiting

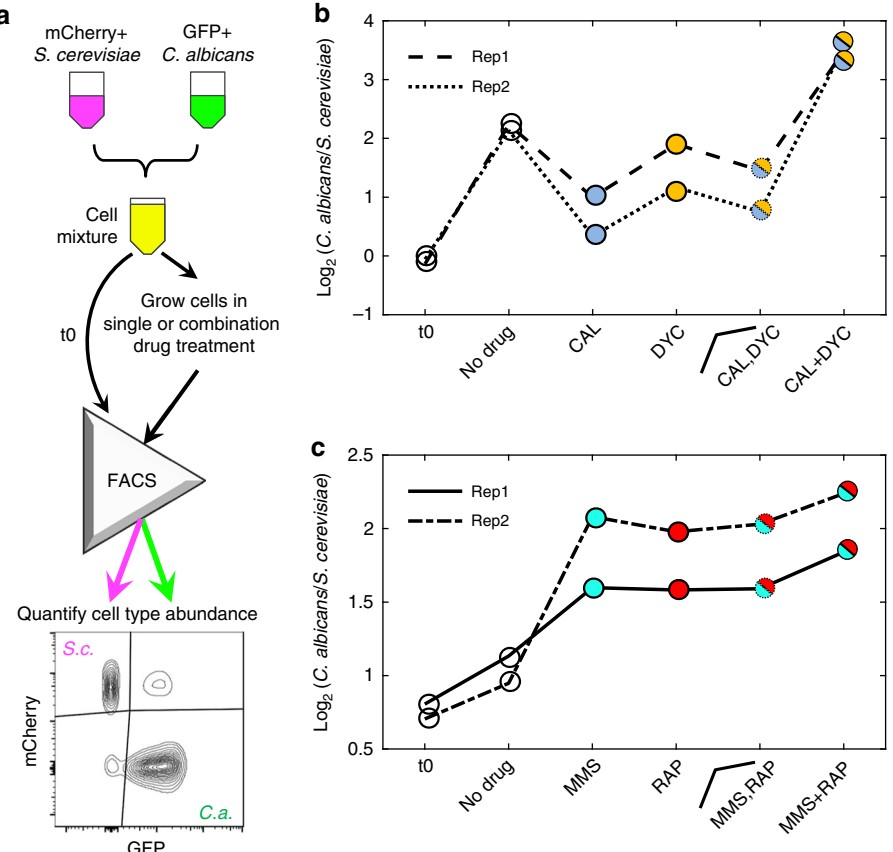

**Fig. 4** Co-culture experiments validate the selectivity model predictions. **a** mCherry expressing *S. cerevisiae* and GFP expressing *C. albicans* cells were co-cultured in single or combination drug treatments in liquid media and growth of each species was quantified using flow cytometry. **b** Selectivity scores (log$_2$(*C. albicans*/*S. cerevisiae*)) for mixed culture assays before treatment (t0), at no drug condition, CAL, DYC, and CAL + DYC co-culture experiments are shown ($n = 2$). Also shown is the average selectivity from CAL and DYC conditions, which is the expected selectivity in the absence of drug interactions. Comparison of the no drug condition to t0 shows that the amount of *C. albicans* in co-culture increases without any selective pressure, which is expected due to the shorter doubling time of *C. albicans*. Comparison of CAL and DYC to no drug condition validates the model prediction of single-drug selectivity for *S. cerevisiae*. Comparison of CAL + DYC to no drug condition indicates that the combination is selective for *C. albicans*, as predicted by the selectivity model (inverted selectivity). **c** MMS and RAP both individually select for *C. albicans*. As predicted by the model, the MMS + RAP combination has greater selectivity for *C. albicans* (selectivity increase due to antagonism). ($n = 2$)

drugs[12], the same is also true of antagonistic combinations, because it is the difference in drug interactions between cell types that enhances or diminishes the therapeutic window. Here, we provided a proof-of-concept that drug interactions may shift selectivity with respect to single-drug effects in mixed microbial communities. Flow cytometry assessment of mixed yeast cultures illustrated that a strong synergistic interaction between calyculin A and dyclonine in *S. cerevisiae* selected for the growth of *C. albicans*, as expected. However, for the combination of MMS and rapamycin, the strength of antagonism selected for *C. albicans*, both in growth and survival assays. Importantly, synergy does not guarantee enhanced selectivity, with synergistic "off-target" effects capable of diminishing or even inverting the therapeutic selectivity.

We found that synergistic drug interactions for the 12 antifungals tested were significantly conserved between these two yeast species, while antagonistic interactions were not conserved. A likely explanation for this is promiscuous synergy in which one drug can affect the bioavailability of many other drugs, e.g., via effects on membrane composition. Indeed, it seems likely that much of the synergy for drugs targeting ergosterol biosynthesis in this study (DYC, FEN, HAL, TER) is due to increased bioavailability of partner drugs. Pentamidine has also been previously identified as a promiscuously synergistic drug[10], although the

mechanisms underlying this promiscuity remain unknown. By contrast, only 3 of the 12 antifungals (BEN, BRO, STA) from our panel have previously been identified as frequently participating in antagonistic interactions[19].

We used the checkerboard assay for a full appreciation of interaction and selectivity as a proof-of-principle and found that selectivity scores at $\theta = 45$ are significantly correlated with selectivity scores at $\theta = 23$ and $\theta = 66$. This indicates that a simplified method for determining selectivity for equi-inhibitory quantities of two drugs may provide a useful approximation of the selectivity of drug combinations[20–22].

Methods such as multiplex ELISA, PCR, and gene sequencing allow cost-effective experiments. Drug-interaction assays are generally conducted using a single microbe type or cell line. With the co-culture method we described, the interactions for more than one species can be measured in one experiment. Our study uses a multiplexed drug-interaction assay, where the interaction is simultaneously determined for multiple species in a heterogeneous culture. We propose that this approach could be applied to mixed cultures of cancer cell lines tagged by DNA barcodes[23] in order to efficiently identify drug combinations with selective synergy against specific cancer genotypes.

Though differential drug interactions are common and, we propose, important in the design of combinations, they were

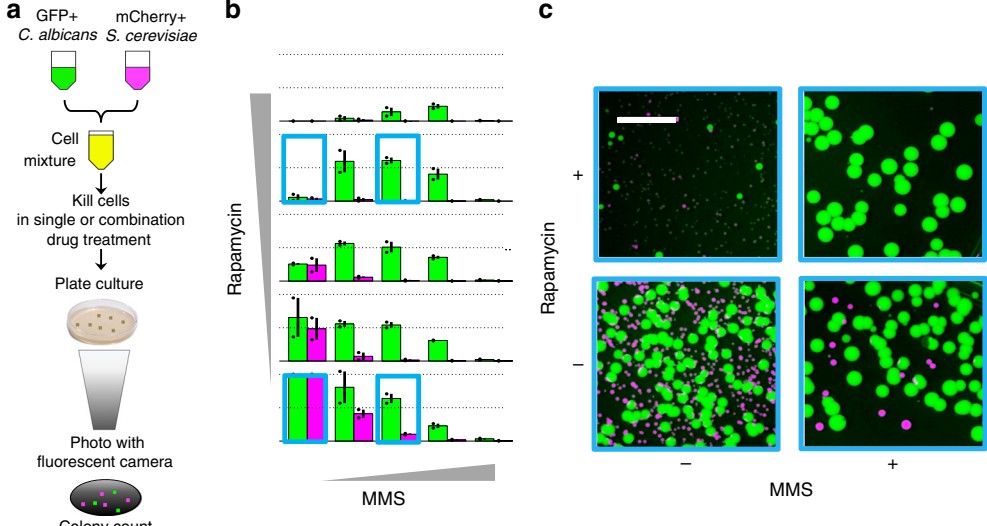

**Fig. 5** Multiplexed fungicidal drug-interaction assay illustrates selectivity increase via antagonism. **a** mCherry expressing *S. cerevisiae* and GFP expressing *C. albicans* cells were co-cultured in 2D grids of drug combinations in liquid media and transferred to YPD-agar plates. After 48-hour incubation, plates were photographed and cells enumerated with ImageJ software. **b** Bar charts of size proportional to cell number compared to the no drug control and color representative of species (green: *C. albicans*, magenta: *S. cerevisiae*) are shown for each MMS-rapamycin combination tested. For each subplot, the top dashed line represents CFUs equal to those observed for the no drug control and the second dashed line represents half the CFUs relative to those observed for the control. Error bars represent ± S.E.M. of two independent experiment results, overlaid on the bar charts as dots (*n* = 2). The experiments indicated with boxes correspond to four representative images of colonies post-incubation in variable drug conditions shown in **c**. **c** While rapamycin and MMS are both toxic for these two yeast species, a low MMS-high rapamycin combination selects for the survival of *C. albicans*. The gray line is 15 mm, all images are of the same size/scale

observed against an overall strong conservation of drug interactions between two species separated by hundreds of millions of years of evolution[24]. Thus, these results provide a strong rationale for screening drug interactions in model organisms or cell lines to prioritize promising combinations for testing in related pathogens. We also predict that drug combinations that are synergistic against a drug-sensitive cell type may remain therapeutically relevant against drug-resistant strains whose genetic similarity remains high. This prediction is in agreement with a recent study that showed that gene deletions rarely lead to a change in drug interactions among *E. coli* non-essential gene deletion strains[25]. Similar reasoning predicts that drug combinations that are synergistic against cancer cells may be accompanied with synergistic toxic side effects given the inherent similarity between cancer cells and normal cells from the same patient. While the discovery of synergistic anti-cancer drug combinations is a growing area of research, the therapeutic potential of synergistic drug pairs must be pursued with caution considering the possibility of enhanced toxicity. For example, the use of combination immunotherapy in melanoma increases the objective response rate by 14% but at the cost of more than doubling the rate of toxic effects so severe that 36% of the patients discontinue treatment[26].

We note that while our co-culture assay is an exciting means to detect selectivity in heterogeneous cell cultures, our analytical model is also applicable to other clinically relevant selectivity considerations. Multiplexed drug-interaction assays may be employed to detect selectivity under very specific conditions, for example, commensal vs. pathogenic microbes that that may be cultured together. These assays are especially useful to simultaneously measure drug interactions for drug resistant and sensitive microbes in order to identify concentration regimes of the two-drug space that specifically select against drug-resistant strains[15]. However, in many therapies, toxicity occurs at tissues that are not at the affected site of infection or disease, and are therefore not amenable for co-culture. For example, ototoxicity arising from

aminoglycoside antibiotics[27]; nephrotoxicity arising from vancomycin, aminoglycosides, and some beta-lactams[28,29]; cardiotoxicity from the chemotherapy doxorubicin and trastuzumab[30]; or peripheral neuropathy associated with chemotherapies including cisplatin, vincristine, and paclitaxel are all clinically observed drug toxicities that could be modeled by this framework of selectivity[31].

By superimposing two systematic drug-interaction experiments in distantly related yeast species, we generated a framework for measuring selectivity of individual drugs and their combinations. The analysis developed for this study provides a model for assessing drug efficacy vs. side effects for combinations. This strategy has immediate applications to the evaluation of therapeutic potential of combination therapies or predicting adverse side effects. Further studies may assess selectivity of drug combinations in cancer vs. normal cells to limit toxicity, or pathogenic vs. commensal microbes (e.g., *S. aureus* vs. *S. epidermidis*) to preserve the microbiome under antibiotic treatment.

## Methods

**Drug-interaction assessment.** BEN, BRO, FEN, HAL, MMS, PEN, and TER were purchased from Sigma-Aldrich. CAL, RAP, STA, TAC, and TUN were purchased from AG Scientific; DYC was purchased from Toronto Research Chemicals (Table 1). All drugs were dissolved in DMSO or water and stored at −20 °C. For *C. albicans* strain SC5314, yeast cells were grown in YPD (1% yeast extract, 2% bacto-peptone, 2% glucose) overnight and diluted to an $OD_{600}$ of 0.1 in YPD with the desired solvent concentration controlled for final solvent concentration of 2% DMSO at 30 °C. Yeast cells were grown in liquid culture in an 8 × 8 grid on 96-well plates with linearly increasing quantities of drug on each axis from zero drug to approximately minimal inhibitory concentration. Plates were incubated for 16 h in a Tecan Genios microplate reader; with $OD_{595}$ readings every 15 min. Additional drug-interaction assays (MMS tested against MMS, BRO, CAL, DYC, FEN, HAL, RAP, and TUN), in *S. cerevisiae* strain BY4741 were conducted using the same setup as described for *C. albicans*, with a duration of 24 h. The raw data for these experiments are provided as a Supplementary Data 1 (also available at (https://doi.org/10.6084/m9.figshare.6849068)). We used the area under $OD_{595}$ curve of each condition as a metric of cell growth, and standardized growth level to the drug-free condition. Alternative growth metrics such as slope of growth curve and end-point

OD strongly correlated with area under growth curve (Supplementary Fig. 10). Drug-interaction scores obtained using variable growth metrics also strongly correlated (Supplementary Fig. 11).

A drug-interaction score ($\alpha$) was defined by assessing the concavity of the longest isophenotypic contour in the drug-interaction grid[10]. The Loewe additivity model for drug interactions shows that isophenotypic contours are straight lines ($\alpha = 0$) for a drug "combined" with itself (a "self–self" combination), which serves as the reference that defines non-interacting, or "additive" combinations[6,32].

**Selectivity assessment**. To assess selectivity of drug combinations for a specific yeast strain, isophenotypic curves at the greatest level of inhibition observed in both species are superimposed on a drug-interaction grid adjusted for individual strain concentration-response. Linear interpolation of the area under the growth ($OD_{595}$) curve was used to identify common inhibitory levels in the $8 \times 8$ checkerboard of drug response. Considering drug-interaction contour plots in polar coordinates, an angle reflects the relative fraction of each drug within a pair: as $\theta$ changes from 0 to 90 degrees, the fraction of drug A increases and the fraction of drug B decreases. The x- ($\theta = 0$) and y- ($\theta = 90$) intercepts represent the relative inhibitory concentration of drug B and A alone. The distance (d) from the origin of each point along the isophenotypic curve for each species at each angle represents the relative amount of each drug combination to achieve the selected level of inhibition. The selectivity score for *C. albicans* compared to *S. cerevisiae* is defined as $\log_2(d_{albicans}/d_{cerevisiae})$ for $\theta = 45$. Selectivity scores at $\theta = 45$ significantly correlated with selectivity scores at $\theta = 23$ and $\theta = 66$ (Spearman correlation test, $r = 0.95$, $r = 0.94$, respectively; Supplementary Fig. 12) and we therefore used selectivity at $\theta = 45$ for further comparisons of selectivity. Selectivity scores obtained under variable growth metrics also strongly correlated (Supplementary Fig. 13). To account for selectivity discrepancies between single drugs in a combination, we computed an expected selectivity metric assuming additive interactions between drugs in both species. Expected selectivity was assessed by connecting each species' set of x- and y- intercepts with a straight line and computing a selectivity score based on the relative distance from the origin to each contour for $\theta = 45$.

**Co-culture combination treatment assay and flow cytometry**. *S. cerevisiae* (mCherry) and *C. albicans* (GFP) were grown in YPD liquid culture overnight at 30 °C to $OD_{600} = 0.5$, diluted to $OD_{600} = 0.1$, and combined in approximately equal number of cells based on flow cytometry. Cells were then co-incubated on 96-well plates in no drug, single drugs or 1:1 ratio combination of drugs. Final concentrations in single-drug conditions were as follows: (CAL) = 4 µg/ml, (DYC) = 500 µg/ml, (MMS) = 50 µg/ml, (RAP) = 1 ng/ml. Each well had a final volume of 160 µl with a solvent concentration of 2% DMSO. Cells were incubated for 4 h shaking at 900 rpm, at 30 °C. This incubation period maintains culture heterogeneity as the proliferation rates as *C. albicans* doubling time is shorter than *S. cerevisiae* (2 h vs 2.5 h). Strains with similar growth rates may be amenable to co-culture for longer duration as in previous studies[33]. Cell/drug mixtures were assessed for the relative abundance of each yeast species by flow cytometry. For all experimental conditions, >20,000 events were acquired using an Attune NxT Flow cytometer. Events were gated by forward and side scatter, and fluorescence distributions were calculated in FlowJo. Single cell cultures were used to define the gates for GFP+ and mCherry+ yeasts, representing *C. albicans* and *S. cerevisiae*, respectively.

**Multiplexed fungicidal drug-interaction assay**. *S. cerevisiae* (mCherry) and *C. albicans* (GFP) were grown in YPD liquid culture overnight at 30 °C, diluted to $OD_{600} = 0.02$, and combined in equal volume and cell density (CFU/ml). These yeasts had similar growth rates and concentration-responses to unlabeled strains. Cells were then co-incubated on 96-well plates in $5 \times 5$ grid with two-fold serial dilutions of MMS (max concentration, 2 mg/ml) and RAP (max concentration, 1 µg/mL) on each axis, including a zero-drug condition. Each well had a final volume of 100 µl with a solvent concentration of 2% DMSO. Cells were incubated for 1 h shaking at 600 rpm, at 30 °C in a ThermoFisher microplate shaker. Cell/drug mixtures were then diluted 1/10 in YPD and 50 µl of diluted cells from each condition were transferred to individual YPD-agar plates for enumeration. After 48 h of incubation at 30 °C, plates were photographed with a custom-built fluorescence imaging "Macrosope" device[34] to visualize bright-field (1/10), GFP (0.4"), and mCherry (3.2") with aperture 5.6 and ISO 100. The colonies were then enumerated using ImageJ colony counter (size: 400–6000 pixels$^2$, circularity: 0.85–1).

**Cell lines**. Wild-type *C. albicans* and *S. cerevisiae* were purchased from ATCC. Fluorescent *C. albicans* and *S. cerevisiae* were kindly provided by the Cowen Lab of University of Toronto and the Springer Lab of Harvard Medical School, respectively.

**Code availability**. Codes to generate interaction and selectivity metrics are available upon request.

**Data availability**. The data that support the findings of this study are available from the corresponding author upon request. Newly conducted drug-interaction assays and growth assays are available at (https://doi.org/10.6084/m9.figshare.6849068).

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

## Acknowledgements

We thank the Laboratory of Systems Pharmacology at Harvard Medical School and the Boone lab at University of Toronto for providing access to their equipment for experiments, the Cowen lab for providing GFP-*albicans*, the Springer lab for providing mCherry-*cerevisiae*. We thank Michael Baym, Remy Chait, and Roy Kishony for providing access and support in use of fluorescence photography. The authors would like to thank Susan E Leeman for critical discussion of the manuscript. Z.B.W. was supported by NIGMS Training Program in Biomolecular Pharmacology T32GM008541. A.C.P. was supported by NHMRC Early Career Fellowship 1072965 and NIH grant GM107618. P.A.C. was supported by the US National Cancer Institute's Cancer Target Discovery and Development (CTD²) Network (grant number U01CA176152). A.S.K. was supported by an NSF CAREER Award (MCB-1350949), an NIH New Innovator Award (1DP2AI131083-01), and an Individual Biomedical Research Award from The Hartwell Foundation. F.P.R. was supported by the Canada Excellence Research Chairs Program, and by NIH grant HG004233. M.C. was supported by NIGMS Grant P50GM107618, TUBITAK Grant 115S934, and the Turkish Academy of Sciences GEBIP Programme.

## Author contributions

Z.B.W., A.S.K., P.A.C., F.P.R., and M.C. designed the study. Z.B.W., S.K., and M.C. conducted the experiments. Z.B.W., N.K., and M.C. analyzed the data. Z.B.W., A.C.P., M.H.Z., F.P.R., and M.C. wrote the paper.

## Additional information

**Competing interests:** The authors declare no competing interests.

