## [Peer Review file · Nature Communications]

Reviewers' comments:

Reviewer #1 (Remarks to the Author):

Conceptually, this work by Weinstein and co-authors is important since it highlights the significance of cell-type specific drug interactions. While this general issue is known, it has not been considered in great detail in the recent literature. The notion that synergism between drugs may not be helpful unless it can differentiate between the pathogen and the host, has often been neglected. This paper correctly points out that in the case of combinatorial treatment of e.g. eukaryotic pathogens or cancer, synergism itself does not guarantee a larger therapeutic window, and thus synergism selective to the pathogen/malignant cells needs to be looked for.

This paper lays the foundation for filling this gap by identifying drug-drug interactions that differ strongly between *S. cerevisiae* and *C. albicans*. The authors show that both synergism and antagonism of drug pairs can be cell type specific, and that both selective synergism or selective antagonism can increase, decrease, or even invert the therapeutic window. The authors performed drug interaction measurements of 8 drug pairs for *S. cerevisiae* and 66 drug pairs for *C. albicans*; they also made use of data published in Cokol 2011. The main finding of the paper is the demonstration that drug pairs can exhibit quantitatively and, in some cases, qualitatively different drug interactions in as closely related species as *S. cerevisiae* and *C. albicans*. This observation suggests that such differentially interacting drug combinations might be found even for closely related cell types such as human malignant and non-malignant tissue.

While this work is overall certainly interesting, there are technical aspects that require attention and my major concern is that the data to support the main finding is quite limited in the current version (see major concerns below). The manuscript could be considered for publication if the authors can address these concerns.

Major concerns:

* The authors find a striking example of inversion of therapeutic window for the DYC-CAL drug pair: while each of the drugs separately are selective against *C. albicans*, together they are selective against *S. cerevisiae*. This could be an important example of how combination treatment could be effective against pathogens that the drugs alone cannot combat; further, if considered the other way round, it provides a cautioning tale against using untested drug combinations. However, for reasons that do not become clear, the authors do not validate this finding in Fig. 4, but instead chose to validate the interaction between Rapamycin and MMS. They substantiate this choice as an example of a 'counter-intuitive prediction' -- namely that the two drugs antagonize in both species, but due to the strength of this antagonism they favour one of those species in combination more than they do alone. It remains unclear what the authors consider counter-intuitive about this phenomenon. I would be important to clarify this point and validate the most striking predictions of their analysis for other drug pairs in experiments as in Fig. 4.

* The authors suggest that the results of the two drug interaction screens for *C. albicans* and *S. cerevisiae* could predict the outcome of a co-culture of the two species in presence of the drugs. However, for the quantification of growth in the drug interaction screens, the 'area under the growth curve' (AUGC) is used. This measure is somewhat problematic for this purpose since it confounds the effects of lag phase, exponential growth rate, and yield into one metric. For example, a culture with high exponential growth rate but low yield could have the same AUGC value as a culture with low growth rate and high yield. However, in co-culture, the species almost certainly compete for resources; hence, the species with higher growth rate would be expected to prevail. The authors should re-analyse their existing data and quantify the exponential growth rate (beside the AUGC), show whether the choice of the metric influences the interaction score, and evaluate how well each metric (AUGC, exponential growth rate, lag phase) predicts the outcome of the co-culture experiment. Additional measurements of co-culture for several drug pairs would be

crucial to corroborate that this approach indeed makes correct predictions (see below).

*Closely related to the previous point: the authors score the interaction based on AUGC, which is a measure of growth; however, they validate their result by counting CFU (Fig. 4), which measures the number of viable cells. Some analyses showing how these quantities relate to each other for various drug pairs would benefit the paper. The authors could also consider using flow cytometry as a method for analysing growth rather than viability. Flow cytometry would also allow higher throughput use, thus enabling them to address the issues mentioned in the previous point in a relatively simple way.

*The authors validate their findings for a single drug pair. From this analysis it is unclear how general the conclusion really is. A single example as currently shown in Fig. 4 may agree with the authors' prediction by chance. It should be relatively straightforward to perform these co-culture experiments for a larger set of drug pairs representing the different phenomena shown in Fig. 3.

* In Fig. 4, the authors validate a quantitative finding: both single drugs alone favour *C. albicans*, and in combination they favour *C. albicans* even more. Fig 4C, however, makes a qualitative point and does not really show that this quantitative prediction is validated: at the chosen concentration of rapamycin, *S. cerevisiae* apparently does not grow in either RAP or RAP+MMS. The data in this panel could in principle be a result of differential MIC, rather than that of a differential interaction. Showing images for the second row of Panel 4B rather than the first one would better serve the purpose. Furthermore, the y-axis of the plot in 4B is unclear (there should be numbers on the axis and error estimates since this is about a quantitative validation). Likewise, plotting the ratio of the two species in a separate plot could be useful.

Minor points:

* In Figure 1b, authors reuse data from Cokol 2011. This should be clearly indicated in the main text section Results and in the Figure Caption.

* Figure 1b is confusing due to the choice of ordering of the drugs along the matrix. It would help if the drug ordering for *S. cerevisiae* would mirror the ordering for *C. albicans*, rather than being related by rotational symmetry; alternatively, the authors could consider showing the data for the two species as two matrices side by side. Furthermore, some data is not shown, potentially due to an error, e.g. Pen x Pen or Cal x Cal for *C. albicans* or MMS x MMS for *S. cerevisiae* -- this needs to be corrected.

* The words 'selective for' are used in a confusing way. When the authors say that a drug combination is selective for *C. albicans*, they rather mean that the drug combination selectively kills *S. cerevisiae* as opposed to *C. albicans*. My understanding of the expression 'drug selectivity' is that e.g. highly selective cancer drugs kill cancer very selectively, and that by analogy, highly selective drugs against *C. albicans* kill *C. albicans* highly selectively. I would suggest the authors replace the use of 'selective for' with 'selective against' and adjust the selectivity score with respect to killing the pathogenic species, i.e. *C. albicans* such that higher selectivity score means higher relative killing of *C. albicans* compared to *S. cerevisiae* (it is vice versa at the moment).

* Several discussion points would need slight reformulation:

- The authors mention at several points throughout the manuscript that it is surprising that a species-specific antagonistic interaction (as opposed to synergistic interaction) can also enhance selectivity. It is not clear why this is surprising.
- It is not clear what the authors mean by 'multiplexed' in 'first demonstration of multiplexed drug interaction'. It is also unclear if the authors are saying that no one has studied drug interactions in co-culture of various strains before.
- 'Thus, these results provide a strong rationale for screening drug interactions in model organisms or cell lines to prioritize promising combinations for testing in related pathogens.' It is unclear

whether it is not more beneficial to actually screen directly the co-culture of species of interest rather than screen them in separation. One could make a stronger case for studying them in separation if one could demonstrate that in general, these results can be reliably combined to predict the outcome of co-culture (see one of the Major Issues above).

Reviewer #2 (Remarks to the Author):

The manuscript of Weinstein et al. investigates how differential drug interactions between cell types can enhance or diminish the selective killing of the target cell type. The key finding of the paper is that differential drug interactions can commonly influence the cell-type specificity of drug combinations. By focusing on 66 pairwise combinations of 12 antifungal drugs in *C. albicans* and *S. cerevisiae*, the authors found 41 out of 66 combinations to show a significant difference in drug interaction from additive expectation based on self-self drug pairs' experimental variance. The authors showed that drug interactions can often influence the cell-type specificity of drug combinations, with the effect of i) enhancing, ii) diminishing, or iii) inverting the selective killing of the target cell type. The authors also claim that while synergistic combinations can indeed increase the cell-type selectivity of growth-inhibiting drugs, the same is also true of antagonistic combinations, since it is the difference in drug interactions between cell types that enhances or diminishes the therapeutic window. For the first time the authors also demonstrated cell type selectivity in a multiplex interaction assay, where the interaction was simultaneously determined for multiple cell types in a heterogeneous culture.

The concept of therapeutically relevant selectivity of drug combination approaches is not novel (Lehar et al. 2009, Bulusu et al. 2016, Baym et al 2016) and, it also remains unclear for me whether this work can really offer considerably novel framework for measuring selectivity of drug combinations compared to a recent investigation (Lehar et al. 2009). Moreover, I feel that their actual experimental dataset lacks clinical relevance. The authors tested the therapeutic selectivity in a model system representing two yeast species (*C. albicans* and *S. cerevisiae*) rather than using human cell line proliferation assays as toxicity models. However, they claim that the analysis that they have developed for this study provides a novel model for assessing drug efficacy versus side effects for combinations at varying concentration ratios for both antimicrobial and anticancer drugs.

Specific comments:

1. The description of selectivity assessment of drug combination is not clear in either the main text or in the methods section. I think a more detailed explanation and a separate supplementary figure explaining it more clearly would be warranted for the reader to understand.
2. In order to assess selectivity of drug combinations did the authors compare the isophenotypic curves at the same inhibition level of both species? Was it also comparable among the drug pairs?
3. Did the authors use fitted data or experimentally measured data to calculate the selectivity score?
4. It would be important to provide some error estimates of the selectivity scores based on replicate experiments as the authors do not always have many experimentally measured data points on the fixed dose ratio line ($\theta=45$ diagonal frame) (see for example on Figure 1 between Fen-Cal, where there was no observed cell growth in the combination in both of the cell types).
5. The selectivity score was defined using the intercepts ($\theta=45$) on the isophenotypic curves at the greatest level of inhibition. In order to get this score faster and more accurately, did the authors really need to measure the whole drug combination matrix (8x8 grid on 96-well plates with linearly increasing quantities of drug)? Wouldn't mixing the two drugs in equal effective ratio (e.g. 1xMIC from drugA and 1xMIC from drugB) and comparing the observed MIC of this drug mixture in the

two different cell types lead to similar quality and quantity of information?

6. In their model system the MICs of the drugs were mainly comparable between the two yeast species (only 2-10 fold differences). Do the authors have any approximation on how much would the differential drug interaction influence the cell type specificity if the MIC of one or both of the drugs would be 10^2 or 10^3 fold higher in one of the cell types than in the other? This could actually often be the case when considering cells resistant to either one of the drugs in combination.

7. Do the authors have any explanation why synergistic, but not antagonistic, interactions significantly overlapped in these related species? One would expect that synergism occurs in a narrower range of cell types, since both drug targets need to be present and expressed for the synergism to occur, and not all cell types share similar expression patterns for potential target genes.

8) The title of the supplementary figure 1B is: "Drug combinations have a greater range of selectivity than single agents." However, the range of the single agent selectivity score is missing on this figure. Could you please clarify this statement?

Minor points:

1) The method section suggests that on Fig 3 the x and y axes represent relative and not exact concentrations. Could the authors specify this on the figure and also include some values on the axes (most importantly where the axis is equal to 1).

2. In Fig 4(b), what does y-axis mean, CFU? Is it in log-scale or linear scale?

3. Please indicate on Figure 4D which are the 4 concentration combinations depicted from the 5x5 matrix.

4. Please specify the selectivity on Supplementary Figure 1B,C and D? Does it mean the calculated selectivity score ($\log_2(d(\text{albicans})/d(\text{cerevisiae}))$) for $\theta = 45$?

Reviewers' comments:

Reviewer #1 (Remarks to the Author):

Conceptually, this work by Weinstein and co-authors is important since it highlights the significance of cell-type specific drug interactions. While this general issue is known, it has not been considered in great detail in the recent literature. The notion that synergism between drugs may not be helpful unless it can differentiate between the pathogen and the host, has often been neglected. This paper correctly points out that in the case of combinatorial treatment of e.g. eukaryotic pathogens or cancer, synergism itself does not guarantee a larger therapeutic window, and thus synergism selective to the pathogen/malignant cells needs to be looked for.

This paper lays the foundation for filling this gap by identifying drug-drug interactions that differ strongly between *S. cerevisiae* and *C. albicans*. The authors show that both synergism and antagonism of drug pairs can be cell type specific, and that both selective synergism or selective antagonism can increase, decrease, or even invert the therapeutic window. The authors performed drug interaction measurements of 8 drug pairs for *S. cerevisiae* and 66 drug pairs for *C. albicans*; they also made use of data published in Cokol 2011. The main finding of the paper is the demonstration that drug pairs can exhibit quantitatively and, in some cases, qualitatively different drug interactions in as closely related species as *S. cerevisiae* and *C. albicans*. This observation suggests that such differentially interacting drug combinations might be found even for closely related cell types such as human malignant and non-malignant tissue.

While this work is overall certainly interesting, there are technical aspects that require attention and my major concern is that the data to support the main finding is quite limited in the current version (see major concerns below). The manuscript could be considered for publication if the authors can address these concerns.

We thank the reviewer for their positive comments, and have endeavored to address the concerns.

Major concerns:

1. The authors find a striking example of inversion of therapeutic window for the DYC-CAL drug pair: while each of the drugs separately are selective against *C. albicans*, together they are selective against *S. cerevisiae*. This could be an important example of how combination

treatment could be effective against pathogens that the drugs alone cannot combat; further, if considered the other way round, it provides a cautioning tale against using untested drug combinations. However, for reasons that do not become clear, the authors do not validate this finding in Fig. 4, but instead chose to validate the interaction between Rapamycin and MMS. They substantiate this choice as an example of a 'counter-intuitive prediction' -- namely that the two drugs antagonize in both species, but due to the strength of this antagonism they favour one of those species in combination more than they do alone. It remains unclear what the authors consider counter-intuitive about this phenomenon. I would be important to clarify this point and validate the most striking predictions of their analysis for other drug pairs in experiments as in Fig. 4.

We thank the reviewer for these thoughtful suggestions. The revised manuscript now clarifies the technical and conceptual reasoning behind pursuing the MMS + Rapamycin pair in the co-culture analysis of selectivity. *Technical*: the selective killing experiment in Figure 4 is only appropriate for rapidly fungicidal (cell-killing) drugs. Only MMS + Rapamycin display this property, while the other compounds are merely fungistatic (growth-inhibiting). These distinctions between fungicidal and fungistatic effects are now demonstrated in Supplementary Figure 6. *Conceptual*: whereas previous studies have sought selectivity arising from synergy vs. non-synergy (as in Lehar et al., 2009), we felt that selectivity increase due to increased antagonism in one species was underappreciated and thus worthy of further investigation. We agree with the reviewer that 'counter-intuitive' is not the right word choice: the revised manuscript describes this interaction as being in an under-appreciated class of interactions that alter selectivity due to changes in interaction strength. To avoid the suggestion that all other pairs are in any sense un-validated, we now describe the co-culture analysis of MMS + Rapamycin as an illustration of selectivity rather than a validation.

We note that while our co-culture assay is an exciting means to detect selectivity, our analytical model is more broadly applicable to clinically relevant selectivity than specific microbes that may be cultured simultaneously. Our study uses *C. albicans* and *S. cerevisiae* as a model to explore the broad clinical issue of selectivity of drug combinations between intended and unintended drug effects. For many therapies, unwanted toxicity occurs at tissues that are not co-located at the site of infection or disease. For these cases, a mixed-culture experiment lacks biological relevance, and selectivity is most appropriately measured with drug interaction experiments for each cell type grown separately (as constitutes most of the data in this study). Among cancer therapies the great majority of dose-limiting toxicities occur in tissues distant to a cancer, and examples of approved drug combinations with overlapping toxicity include the cardiotoxicity of doxorubicin and tyrosine kinase inhibitors, and myelosuppression by

nucleoside analogs and platinum chemotherapies. Among antimicrobial therapies, examples include nephrotoxicity from vancomycin, beta-lactams, and aminoglycosides; and ototoxicity from aminoglycosides and macrolides. For example, nephrotoxicity is a significant concern with antibiotics even when not treating a kidney infection. For such cases, it would not be therapeutically relevant to co-culture kidney cells with bacteria. For this reason, we believe that experiments in separate cultures of *C. albicans* and *S. cerevisiae* are sufficient to prove the existence of differential drug interactions and their resulting impact on cell-type selectivity. The revised manuscript now discusses the important issues raised above about toxicity in tissues that are not at the site of infection or disease, where co-culture is neither biologically appropriate nor experimentally tractable.

2. The authors suggest that the results of the two drug interaction screens for *C. albicans* and *S. cerevisiae* could predict the outcome of a co-culture of the two species in presence of the drugs. However, for the quantification of growth in the drug interaction screens, the 'area under the growth curve' (AUGC) is used. This measure is somewhat problematic for this purpose since it confounds the effects of lag phase, exponential growth rate, and yield into one metric. For example, a culture with high exponential growth rate but low yield could have the same AUGC value as a culture with low growth rate and high yield. However, in co-culture, the species almost certainly compete for resources; hence, the species with higher growth rate would be expected to prevail. The authors should re-analyse their existing data and quantify the exponential growth rate (beside the AUGC), show whether the choice of the metric influences the interaction score, and evaluate how well each metric (AUGC, exponential growth rate, lag phase) predicts the outcome of the co-culture experiment. Additional measurements of co-culture for several drug pairs would be crucial to corroborate that this approach indeed makes correct predictions (see below).

Following the reviewer's suggestion, we reanalyzed our data using the maximum slope and the end-point OD of each growth curve as other proxies for growth. We found that the growth metrics using all 3 methods were significantly correlated (AUGC with max slope and end-point OD, respectively, were highly correlated; $r = 0.96$ $r = 0.95$ for *C. albicans*; $r = 0.96$, $r = 0.96$ for *S. cerevisiae*). Not surprisingly given these high correlations, interaction scores and selectivity measures were also highly correlated ($r > 0.72$ for all interaction and selectivity comparisons by growth metric). AUGC and lag phase were inversely correlated ($r = -0.84$ for *S. cerevisiae*, $r = -0.53$ for *C. albicans*), as expected.

The additional analyses are provided in Supplementary Figures: 9, 10, 12.

We also present our analysis for MMS + Rapamycin, using the maximum slope and end-point OD as growth metrics in Supplementary Figure 7.

3. Closely related to the previous point: the authors score the interaction based on AUGC, which is a measure of growth; however, they validate their result by counting CFU (Fig. 4), which measures the number of viable cells. Some analyses showing how these quantities relate to each other for various drug pairs would benefit the paper. The authors could also consider using flow cytometry as a method for analysing growth rather than viability. Flow cytometry would also allow higher throughput use, thus enabling them to address the issues mentioned in the previous point in a relatively simple way.

The revised manuscript now makes clear that the follow-up study of the MMS + Rapamycin combination was meant as an illustration of selectivity rather than a validation of the original findings. We absolutely agree that measurement of interaction in terms of CFUs would not necessarily replicate the growth-based measures, e.g., for drugs and combinations that are fungistatic rather than fungicidal. We also completely agree that high-throughput application of flow cytometry for the purpose of testing drug interactions would be an exciting direction; but this would represent a substantially new technical direction that could warrant publication of another study. The current manuscript contains over 80 original drug interaction checkerboard experiments representing approximately 5000 experimental conditions that were conducted over months. Establishment and validation of the application of flow cytometry in our hands to measure drug interactions, followed by comprehensive exploration of drug pairs examined already here, would greatly delay the release of results for this study.

4. The authors validate their findings for a single drug pair. From this analysis it is unclear how general the conclusion really is. A single example as currently shown in Fig. 4 may agree with the authors' prediction by chance. It should be relatively straightforward to perform these co-culture experiments for a larger set of drug pairs representing the different phenomena shown in Fig. 3.

As noted above, the revised manuscript now describes the co-culture analysis of MMS + Rapamycin as an illustration of selectivity rather than a validation, avoiding the misleading suggestion that other pairs beyond MMS + Rapamycin are in any sense un-validated. Indeed, the co-culture selectivity experiment was optimized for drugs with fungicidal activity. None of the other 10 drugs had the rapid fungicidal activity that is best-suited for the co-culture assay, so that the co-culture selectivity assay cannot readily be applied to other pairs (Supplementary Figure 6).

5.* In Fig. 4, the authors validate a quantitative finding: both single drugs alone favour *C. albicans*, and in combination they favour *C. albicans* even more. Fig 4C, however, makes a qualitative point and does not really show that this quantitative prediction is validated: at the chosen concentration of rapamycin, *S. cerevisiae* apparently does not grow in either RAP or RAP+MMS. The data in this panel could in principle be a result of differential MIC, rather than that of a differential interaction. Showing images for the second row of Panel 4B rather than the first one would better serve the purpose. Furthermore, the y-axis of the plot in 4B is unclear (there should be numbers on the axis and error estimates since this is about a quantitative validation). Likewise, plotting the ratio of the two species in a separate plot could be useful.

We thank the reviewer for these comments. In our revised submission, we used the second row of Figure 4B for images shown in Figure 4C. We conducted replicate experiments for co-culture experiments and plotted error estimates for bar charts shown in Figure 4B. In addition, we generated a pie chart that represents the ratio of two species provided as Supplementary Figure 8.

Minor points:

6. * In Figure 1b, authors reuse data from Cokol 2011. This should be clearly indicated in the main text section Results and in the Figure Caption.

We regret the oversight. The revised manuscript now indicates the use of previously published data in the results and figure legends as follows:

“Seven *S. cerevisiae* experiments were newly conducted for this study: MMS tested against BRO, CAL, DYC, FEN, HAL, RAP and TUN. Other experimental data involving *S. cerevisiae* were obtained from Cokol et al., 2011”

7. Figure 1b is confusing due to the choice of ordering of the drugs along the matrix. It would help if the drug ordering for *S. cerevisiae* would mirror the ordering for *C. albicans*, rather than being related by rotational symmetry; alternatively, the authors could consider showing the data for the two species as two matrices side by side. Furthermore, some data is not shown, potentially due to an error, e.g. Pen x Pen or Cal x Cal for *C. albicans* or MMS x MMS for *S. cerevisiae* -- this needs to be corrected.

We appreciate the suggestion, and have re-ordered the drugs in Figure 1B as suggested. We also thank the reviewer for leading us to correct and clarify the set of self-self control experiments. As the self-self interactions are conducted to estimate experimental error, we conducted self-self controls for 10 of the 12 drugs in our test set for both yeast species; with the exceptions being Pen-Pen and Cal-Cal. Given the distribution of interaction scores around 0 (mean = -0.01, std. dev. = 0.4), the remaining self-self experiments serve as a model for variability in interaction and selectivity. MMS + MMS for *S. cerevisiae* was missing due to a coding error. In the revised Figure 1B, we plotted the self-self experiments for both species on the diagonal.

8. The words 'selective for' are used in a confusing way. When the authors say that a drug combination is selective for *C. albicans*, they rather mean that the drug combination selectively kills *S. cerevisiae* as opposed to *C. albicans*. My understanding of the expression 'drug selectivity' is that e.g. highly selective cancer drugs kill cancer very selectively, and that by analogy, highly selective drugs against *C. albicans* kill *C. albicans* highly selectively. I would suggest the authors replace the use of 'selective for' with 'selective against' and adjust the selectivity score with respect to killing the pathogenic species, i.e. *C. albicans* such that higher selectivity score means higher relative killing of *C. albicans* compared to *S. cerevisiae* (it is vice versa at the moment).

We should have used the phrase “selects for the growth of” to more clearly indicate evolutionary selection, rather than “selective for” which, as the reviewer correctly notes, will be taken by most to mean “selects against specifically”. We have changed the language throughout to better capture this distinction.

* Several discussion points would need slight reformulation:

- The authors mention at several points throughout the manuscript that it is surprising that a species-specific antagonistic interaction (as opposed to synergistic interaction) can also enhance selectivity. It is not clear why this is surprising.

Following the reviewer's comment, we removed the word 'surprising' in these contexts, noting instead that the phenomenon of species-specific antagonism has been under-appreciated as a source of selectivity.

- It is not clear what the authors mean by 'multiplexed' in 'first demonstration of multiplexed

drug interaction'. It is also unclear if the authors are saying that no one has studied drug interactions in co-culture of various strains before.

We have used the word multiplex to indicate that multiple experiments are being carried out within a single operation. Drug interaction assays are generally conducted using a single microbe type or cell line. With the co-culture method we described, the interactions for more than one species can be measured in one experiment. While we agree that there have been co-culture experiments using resistant vs. sensitive strains of the same species, to the best of our knowledge, there has not yet been a study that has measured more than one drug interaction for 2 species in one 'checkerboard' assay. This is also supported by reviewer 2, who wrote:

“For the first time the authors also demonstrated cell type selectivity in a multiplex interaction assay, where the interaction was simultaneously determined for multiple species in a heterogeneous culture.”

In the revised manuscript, we clarify these points as:

“Methods such as multiplex ELISA, PCR and gene sequencing allow cost-effective experiments. Drug interaction assays are generally conducted using a single microbe type or cell line. With the co-culture method we described, the interactions for more than one species can be measured in one experiment. Our study provides the first demonstration of a multiplexed drug interaction assay, where the interaction is simultaneously determined for multiple species in a heterogeneous culture.”

- 'Thus, these results provide a strong rationale for screening drug interactions in model organisms or cell lines to prioritize promising combinations for testing in related pathogens.' It is unclear whether it is not more beneficial to actually screen directly the co-culture of species of interest rather than screen them in separation. One could make a stronger case for studying them in separation if one could demonstrate that in general, these results can be reliably combined to predict the outcome of co-culture (see one of the Major Issues above).

We thank the reviewer for the opportunity to clarify the quoted sentence. We intended to indicate that, given the overall conservation of drug interactions between *S. cerevisiae* and *C. albicans*, drug synergy screens can be conducted in model organisms to predict interactions in related species.

In the revised manuscript, we also note in the discussion that a limitation of the co-culture assay is its utility in testing multiple microbes or cell types that are amenable to growth and toxicity assays under the same experimental conditions; though the majority of clinically relevant combinatorial drug toxicities may be determined through modeling using individual culture data.

Reviewer #2 (Remarks to the Author):

The manuscript of Weinstein et al. investigates how differential drug interactions between cell types can enhance or diminish the selective killing of the target cell type. The key finding of the paper is that differential drug interactions can commonly influence the cell-type specificity of drug combinations. By focusing on 66 pairwise combinations of 12 antifungal drugs in *C. albicans* and *S. cerevisiae*, the authors found 41 out of 66 combinations to show a significant difference in drug interaction from additive expectation based on self-self drug pairs' experimental variance. The authors showed that drug interactions can often influence the cell-type specificity of drug combinations, with the effect of i) enhancing, ii) diminishing, or iii) inverting the selective killing of the target cell type. The authors also claim that while synergistic combinations can indeed increase the cell-type selectivity of growth-inhibiting drugs, the same is also true of antagonistic combinations, since it is the difference in drug interactions between cell types that enhances or diminishes the therapeutic window. For the first time the authors also demonstrated cell type selectivity in a multiplex interaction assay, where the interaction was simultaneously determined for multiple cell types in a heterogeneous culture. The concept of therapeutically relevant selectivity of drug combination approaches is not novel (Lehar et al. 2009, Bulusu et al. 2016, Baym et al 2016) and, it also remains unclear for me whether this work can really offer considerably novel framework for measuring selectivity of drug combinations compared to a recent investigation (Lehar et al. 2009). Moreover, I feel that their actual experimental dataset lacks clinical relevance. The authors tested the therapeutic selectivity in a model system representing two yeast species (*C. albicans* and *S. cerevisiae*) rather than using human cell line proliferation assays as toxicity models. However, they claim that the analysis that they have developed for this study provides a novel model for assessing drug efficacy versus side effects for combinations at varying concentration ratios for both antimicrobial and anticancer drugs.

We thank the reviewer for these comments. The system of *C. albicans* and *S. cerevisiae* does not itself have direct clinical relevance, but this system has provided an analytical framework with very broad utility to studies of clinically relevant selectivity. The selectivity model

presented in this study can be directly applied to analyze selectivity in cancer cell lines vs. cultures of various normal human tissues (e.g. cardiomyocytes) because the required data is cell growth or death, which can be measured just as readily in human cells as in yeast.

In addition, our analytical model is in no way limited to specific microbes that can be cultured together. For many therapies, unwanted toxicity occurs at tissues that are not co-located at the site of infection or disease. For these cases, selectivity is most appropriately measured with drug interaction experiments for each cell type grown separately, as constitutes most of the data in this study. Among cancer therapies the great majority of dose-limiting toxicities occur in tissues distant to a cancer, and examples of approved drug combinations with overlapping toxicity include the cardiotoxicity of doxorubicin and tyrosine kinase inhibitors, and myelosuppression (bone marrow toxicity) by nucleoside analogs and platinum chemotherapies. Among antimicrobial therapies, examples include nephrotoxicity from vancomycin, beta-lactams, and aminoglycosides; and ototoxicity from aminoglycosides and macrolides.

The references described by the reviewer are familiar and of interest to us and discuss the selectivity of combination therapies. However, in Lehar et al., 2009 and Bulusu et al., 2009, the concept of therapeutically relevant selectivity is discussed exclusively in the context of synergistic interactions, with no suggestion that antagonistic interactions could enhance selectivity. The discovery in this study that the magnitude of difference in interactions (including antagonism) defines the selectivity of drug combinations is a novel contribution. Baym et al., 2016 discusses how drug interactions affect the evolution of drug resistance within a single species, and does not address the consequences of differential drug interactions between cell types or species.

Therefore, we believe that this study presents novel findings that extend beyond the previous work on synergy and selectivity, while providing a model framework that can be broadly applied to other biomedical contexts wherein drug combinations may influence the therapeutic window or model the effects of multiple phenotypes of interest.

Specific comments:

1. The description of selectivity assessment of drug combination is not clear in either the main text or in the methods section. I think a more detailed explanation and a separate supplementary figure explaining it more clearly would be warranted for the reader to understand.

Following the reviewer's comment, we provided a flow diagram as Supplementary Figure 2.

2. In order to assess selectivity of drug combinations did the authors compare the isophenotypic curves at the same inhibition level of both species? Was it also comparable among the drug pairs?

We thank the reviewer for the opportunity to clarify our methods. Yes, we always evaluated the same level of inhibition of isophenotypic curves for both species. These inhibition levels varied among drug pairs with mean 0.31 and standard deviation 0.17. We clarified this in the text as follows:

“We explored the impact of drug interactions on selectivity by superimposing the isophenotypic contours from drug-interaction experiments for each cell type, for the greatest level of inhibition present in interaction datasets for both species (mean inhibitory level = 0.31; std. dev. = 0.17). “

3. Did the authors used fitted data or experimentally measured data to calculate the selectivity score?

The contour generation uses linear interpolation between experimental data points in a checkerboard. We used these contours to calculate selectivity score. We clarified this in the Methods section under ‘Selectivity Assessment’ as follows:

“To assess selectivity of drug combinations for a specific yeast strain, isophenotypic curves at the greatest level of inhibition observed in both species are superimposed on a drug-interaction grid adjusted for individual strain concentration-response. Linear interpolation of the area under the growth (OD_{595}) curve was used to identify common inhibitory levels in the 8x8 checkerboard of drug response.“

4. It would be important to provide some error estimates of the selectivity scores based on replicate experiments as the authors do not always have many experimentally measured data points on the fixed dose ratio line ($\theta=45$ diagonal frame) (see for example on Figure 1 between Fen-Cal, where there was no observed cell growth in the combination in both of the cell types).

Thank you for the opportunity to clarify our rationale for significant drug interaction associated selectivity scores. As the reviewer correctly noted, we use interpolated data to identify shared inhibitory levels for the isophenotypic contour of the checkerboard. In order to overcome this potential limitation, we used the self-self experiments to serve as an error model for significant

selectivity. Significant selectivity was defined as selectivity greater than expected for linear isophenotypic contours. Using this definition, we found that the difference between expected and observed selectivity was zero for self-self controls (lower set of black circles in Supplementary Figure 3). We therefore defined statistically significant selectivity due to drug interactions for drug pairs that had selectivity scores that significantly varied (± 2 standard deviations) from the mean of the self-self controls (mean = 0.12, std. dev. = 0.10). We considered selectivity scores between -0.08 and 0.32 as non-significant based on our error model.

5. The selectivity score was defined using the intercepts ($\theta=45$) on the isophenotypic curves at the greatest level of inhibition. In order to get this score faster and more accurately, did the authors really need to measure the whole drug combination matrix (8x8 grid on 96-well plates with linearly increasing quantities of drug)? Wouldn't mixing the two drugs in equal effective ratio (e.g. 1xMIC from drugA and 1xMIC from drugB) and comparing the observed MIC of this drug mixture in the two different cell types lead to similar quality and quantity of information?

The reviewer makes an excellent observation for a simplified method for determining selectivity for equally inhibitory quantities of two drugs. We used the 8x8 checkerboard assay for a full appreciation of interaction and selectivity as a proof of principle. One drawback of the experimental method suggested by the reviewer is the variation in sampling space of the checkerboard related to daily variations in MIC values. To examine this possibility that sampling different regions of the checkerboard may confound selectivity, we analyzed selectivity metrics for $\theta=23$ and $\theta=66$, representing the midway points between $\theta=45$ and each axis. We found that selectivity scores at $\theta=45$ are significantly correlated with selectivity scores at $\theta=23$ and $\theta=66$. Therefore, we conclude that the methodology suggested by the reviewer could be an excellent means to quickly assess drug interactions and selectivity. We provide this analysis as Supplementary Figure 11, and added the following text:

Methods:

“Selectivity scores at $\theta=45$ are significantly correlated with selectivity scores at $\theta=23$ and $\theta=66$ (Spearman's $r = 0.95$, $r = 0.94$, respectively; Supplementary Figure 11) and therefore used selectivity at $\theta=45$ for further comparisons of selectivity.”

Discussion:

“We used the checkerboard assay for a full appreciation of interaction and selectivity as a proof of principle and found that selectivity scores at $\theta=45$ are significantly correlated with selectivity scores at $\theta=23$ and $\theta=66$. This indicates that a simplified method for determining selectivity for

equi-inhibitory quantities of two drugs (as used by Lehar et al., 2016, and Weinstein & Zaman, 2017 for drug interaction measurement) provides a useful approximation of the selectivity of drug combinations.”

6. In their model system the MICs of the drugs were mainly comparable between the two yeast species (only 2-10 fold differences). Do the authors have any approximation on how much would the differential drug interaction influence the cell type specificity if the MIC of one or both of the drugs would be 10^2 or 10^3 fold higher in one of the cell types than in the other? This could actually often be the case when considering cells resistant to either one of the drugs in combination.

Again, the reviewer suggests an excellent analysis. Indeed, the influence of antimicrobial resistance to therapeutic selectivity is an important consideration in the context of increasing levels of antibiotic resistance worldwide. Following the reviewer’s comment, in the revised submission we provide an analysis where the MIC of one species is multiplied by 100 for one drug; and assess how the change in selectivity score compares with the difference in interaction as **Supplementary Figure 5**. To do this analysis, we assumed that isophenotypic contours scaled with changes in drug sensitivity as previously described by Wood et al., 2014.

<http://labs.mcb.harvard.edu/Cluzel/documents/Cluzel%20Cell%202014.pdf>

In the original submission, we demonstrated that the difference of interaction scores correlate with selectivity scores. Using our resistance analysis, we found that when one of the species evolves 100-fold resistance to one of the drugs in a pairwise combination, the difference of interaction scores no longer correlates with selectivity. Rather, selectivity scores are dominated by the 100-fold resistance, which cannot be overcome by selectivity due to drug interactions. Therefore, we conclude that resistance may strongly influence the selectivity of a drug combination.

We describe our findings in the results section as follows:

“In order to understand the effect of antimicrobial resistance on therapeutic selectivity, we modeled the effects of 100-fold resistance on selectivity metrics for all tested drug pairs. We assumed that isophenotypic contours scaled with changes in drug sensitivity and simulated resistance by multiplying the minimal inhibitory concentration of one compound by 100 while preserving the shape of the drug interaction isobole. We observed that $\Delta\alpha$ and sel_{exp} are not significantly correlated after simulating for resistance, suggesting that extreme drug resistance is more influential on selectivity than variation in drug interactions (Supplementary Figure 5).”

Supplementary Figure 5 Legend:

“There was a weak but significant correlation ($r = 0.26$, $p = 0.02$) between selectivity and the difference of α scores between *C. albicans* and *S. cerevisiae* ($\alpha_{alb} - \alpha_{cer}$) for the tested drug pairs (black circles). In order to understand the influence of antimicrobial resistance on therapeutic selectivity, we modeled the effects of a 100-fold change in minimal inhibitory concentration of one drug on selectivity metrics for each pair in either *S. cerevisiae* (red circles) or *C. albicans* (green). We assumed that isophenotypic contours scaled with changes in drug sensitivity (Wood et al., 2014). We found that a large change in MIC overcame the influence of differential drug interactions in our model system and no significant correlation remained for ($\alpha_{alb} - \alpha_{cer}$) and selectivity scores for simulated resistance in *S. cerevisiae* ($r = 0.09$, $p = 0.45$) or *C. albicans* ($r = 0.07$, $p = 0.58$).”

7. Do the authors have any explanation why synergistic, but not antagonistic, interactions significantly overlapped in these related species? One would expect that synergism occurs in a narrower range of cell types, since both drug targets need to be present and expressed for the synergism to occur, and not all cell types share similar expression patterns for potential target genes.

The synergies shared by these two yeast species are likely due to promiscuous synergy, e.g., where one drug affects the bioavailability of a second drug. This is in contrast with specific synergy, where drugs inhibit two targets on parallel pathways. We added this possible explanation to the discussion as follows:

“We found that synergistic drug interactions for the 12 antifungals tested were significantly conserved between these two yeast species, while antagonistic interactions were not conserved. A likely explanation for this is promiscuous synergy in which one drug can affect the bioavailability of many other drugs, e.g., via effects on membrane composition. Indeed, it seems likely that much of the synergy for drugs targeting ergosterol biosynthesis in this study (DYC, FEN, HAL, TER) is due to increased bioavailability of partner drugs. Pentamidine has also been previously identified as a promiscuously synergistic drug (Cokol et al., 2011), although the mechanisms underlying this promiscuity remain unknown. By contrast, only 3 of the 12 antifungals (BEN, BRO, STA) from our panel have previously been identified as frequently participating in antagonistic interactions (Cokol et al., 2014).”

8) The title of the supplementary figure 1B is: “Drug combinations have a greater range of selectivity than single agents.” However, the range of the single agent selectivity score is missing on this figure. Could you please clarify this statement?

In Supplementary Figure 1b, we demonstrate single agent selectivity due to drug interactions in the lower set of black circles within the superimposed plot of frequency distribution. To increase clarity, we have updated the Supplementary Figure 1b (now Supplementary Figure 3) labels and legends. We updated the figure legend as follows:

“For each drug pair in our study, selectivity – selectivity_{exp} was computed using the equation $\log_2(d(\text{albicans})/d(\text{cerevisiae}))$ for $\theta = 45$. The black circles on the bottom row represents single agent selectivity due to drug interactions, as computed by observed selectivity – selectivity_{exp} values for self-self controls. The difference between observed and expected selectivity for self-self controls are distributed around 0, as expected (range: -0.07 to 0.27). The normal distribution fit to self-self controls is given in black (mean = 0.12, std. dev. = 0.1). In the top row of circles, selectivity – selectivity_{exp} values for non self-self pairs are given. Drug pairs that significantly select for *C. albicans* (green, positive) or *S. cerevisiae* (red, negative) versus additive approximation of expected selectivity were identified based on the 95% confidence interval of the self-self combinations. “

Minor points:

1) The method section suggests that on Fig 3 the x and y axes represent relative and not exact concentrations. Could the authors specify this on the figure and also include some values on the axes (most importantly where the axis is equal to 1).

We updated Figure 3 axes as suggested by the reviewer. We note in the figure legend that x and y represent relative concentrations and the red contour intercept the x- and y- axis at 1 as follows:

“Observed isophenotypic contours of drug-interaction assays for *S. cerevisiae* (red) and *C. albicans* (green) are overlaid in a 2D grid adjusted for relative concentration. We linearly transformed the isophenotypic contours for drug-interaction assays so that *S. cerevisiae*’s isophenotypic contour intercepted both x and y axes at 1.”

2. In Fig 4(b), what does y-axis mean, CFU? Is it in log-scale or linear scale?

We thank the reviewer for pointing out this oversight. The y axis in Figure 4B represents the number of CFUs in the given condition relative to that in the no drug condition and are shown on a linear scale. Two gray horizontal lines correspond to 0.5 and 1. The figure legend is updated as follows to reflect this information:

“Bar charts of size proportional to cell number compared to the no drug control and color representative of species (green: *C. albicans*, red: *S. cerevisiae*) are shown for each MMS + Rapamycin combination tested. For each subplot, the top dashed line represents CFUs equal to those observed for the no drug control and the second dashed line represents half the CFUs relative to those observed for the control. Error bars represent +/- S.E.M. of two independent experiments. The experiments indicated with boxes correspond to 4 representative images of colonies post-incubation in variable drug conditions shown in (c).”

3. Please indicate on Figure 4D which are the 4 concentration combinations depicted from the 5x5 matrix.

We indicated the concentration combinations in Figure 4B for which images are given in Figure 4C with blue boxes. (Selected images have been updated in accordance with comments from Reviewer 1).

4. Please specify the selectivity on Supplementary Figure 1B,C and D? Does it mean the calculated selectivity score ($\log_2(d(\text{albicans})/d(\text{cerevisiae}))$ for $\theta = 45$)?

This is correct. We indicated this in the revised submission Supplementary Figure 1B, C and D (now Supplementary Figures 3,4,5) legends as follows:

“Expected and observed selectivity scores were determined for $\theta = 45$ for Supplementary Figures using the equation $\log_2(d(\text{albicans})/d(\text{cerevisiae}))$.”

Reviewers' comments:

Reviewer #1 (Remarks to the Author):

The revised manuscript by Weinstein and colleagues is considerably improved and many of the issues I had raised are resolved. However, there is still one unresolved major issue:

The authors state in their rebuttal letter that a larger scale co-culture experiment would be largely irrelevant for their results in Fig. 3. ("For many therapies, unwanted toxicity occurs at tissues that are not co-located at the site of infection or disease. For these cases, a mixed-culture experiment lacks biological relevance, and selectivity is most appropriately measured with drug interaction experiments for each cell type grown separately.") This is quite confusing given that the authors use microbes (fungi) in their experiments. These may well be co-localized, e.g. in a polymicrobial infection; a more common scenario is probably that pathogenic microbes are present in the same place as the commensal microbiota. In these situations, one of the most interesting questions seems to be if a drug combination can be used to eliminate the pathogen(s) in a targeted way, without perturbing the microbiota. I agree with the authors that for cancer this situation is likely different but cancer is not studied experimentally in the present work.

Even if we accept the authors' argument, it is unclear why they are performing the co-culture experiment in Fig. 4 at all. This experiment is not applicable to 98% of the investigated drug pairs and thus of limited use as a general method, it does not have a clear relation to the findings shown in Fig. 3 (assay for growth/AUGC vs survival/CFU) and, according to their own argument, it is of little relevance for off-site toxicity. Hence, one could argue that the authors should remove the experiment in Fig. 4 from the article.

However, contrary to the authors' opinion stated in the rebuttal letter, I think that some validation that the experiments done on the two species in separation can predict experimental outcomes in co-culture is certainly necessary to make a stronger point about the relevance of the observed differences in drug interactions. The authors should select their favourite method (it indeed need not be flow cytometry) to measure growth (rather than survival via CFU) and show that what they measure in separation holds in co-culture. This should be done for at least a few drug pairs, including the most striking examples listed in Fig. 3. Without such experiments, their conclusion that the experimental outcome in a system where the two species are competing for the same nutrients is predictable from the experiments done in separation is not justified.

Reviewer #2 (Remarks to the Author):

The revised manuscript is much improved for clarity. The authors have thoroughly addressed all my questions from the initial review. However, I think the authors did not place properly their work in the context of prior studies (Lehar et al. 2009, Bulusu et al., 2009, Baym et al. 2016).

According to the authors, their main findings are the following:

- 1) Providing a model framework that can be broadly applied to other biomedical contexts wherein drug combinations may influence the therapeutic window or model the effects of multiple phenotypes of interest.
- b) their model is broadly applicable because the required data is cell growth or death, which can be measured just as readily in human cells as in yeast.
- a) their analytical model is in no way limited to specific microbes that can be cultured together.

It still remains unclear for me whether their work can really offer a considerably novel model framework for measuring selectivity of drug combinations compared to the recent investigations (Lehar et al. 2009, Bulusu et al., 2009). Is their selectivity score is more accurate or more broadly applicable compared to the ones presented earlier? They claim that their model is more broadly applicable because the required data is cell growth or death, which can be measured just as readily in human cells as in yeast and their model is no way limited to specific microbes that can be cultured together. Are the previous selectivity models not based on cell growth or dead and are they limited to microbes that can be cultured together? If their model is more broadly applicable than the previous models can they provide some sort of evidence for it? It would be important to present for the readership of the Nature Communications what are the real advantages or disadvantages of their selectivity model compared to the ones that are available in the literature.

2) The magnitude of difference in interactions (including antagonism) defines the selectivity of drug combinations is a novel contribution.

I do not think that this is a strong enough claim to be considered as a novelty. Baym et al., 2016 not just discusses how drug interactions can affect the evolution of drug resistance within a single species, but also discusses how can differential drug interactions select against the resistant strain. Baym et al. 2016 also clearly show the concentration regime in the two-drug concentration space that can select against the resistant strain (Baym et al. 2016 Science, Fig 2. Selection inversion approaches and potential strategies.)

I think the most novel aspect of the paper is the demonstration of the multiplexed drug interaction assay. Their study provides the first demonstration of a multiplexed drug interaction assay, where the interaction is simultaneously determined for multiple species in a heterogeneous culture. But, they claim that while their co-culture assay is an exciting means to detect selectivity, their analytical model is more broadly applicable to clinically relevant selectivity than specific microbes that may be cultured simultaneously. However, I believe, that their approach would be very useful to measure drug interaction simultaneously for drug resistant and sensitive microbes in order to identify concentration regimes of the two-drug space that specifically select against drug-resistant strains. In this case, a mixed-culture experiment would be biologically relevant and co-culturing resistant and sensitive strains of the same species would not be a problem.

Reviewers' comments:

Reviewer #1 (Remarks to the Author):

The revised manuscript by Weinstein and colleagues is considerably improved and many of the issues I had raised are resolved. However, there is still one unresolved major issue:

The authors state in their rebuttal letter that a larger scale co-culture experiment would be largely irrelevant for their results in Fig. 3. ("For many therapies, unwanted toxicity occurs at tissues that are not co-located at the site of infection or disease. For these cases, a mixed-culture experiment lacks biological relevance, and selectivity is most appropriately measured with drug interaction experiments for each cell type grown separately.") This is quite confusing given that the authors use microbes (fungi) in their experiments. These may well be co-localized, e.g. in a polymicrobial infection; a more common scenario is probably that pathogenic microbes are present in the same place as the commensal microbiota. In these situations, one of the most interesting questions seems to be if a drug combination can be used to eliminate the pathogen(s) in a targeted way, without perturbing the microbiota. I agree with the authors that for cancer this situation is likely different but cancer is not studied experimentally in the present work.

Even if we accept the authors' argument, it is unclear why they are performing the co-culture experiment in Fig. 4 at all. This experiment is not applicable to 98% of the investigated drug pairs and thus of limited use as a general method, it does not have a clear relation to the findings shown in Fig. 3 (assay for growth/AUGC vs survival/CFU) and, according to their own argument, it is of little relevance for off-site toxicity. Hence, one could argue that the authors should remove the experiment in Fig. 4 from the article.

However, contrary to the authors' opinion stated in the rebuttal letter, I think that some validation that the experiments done on the two species in separation can predict experimental outcomes in co-culture is certainly necessary to make a stronger point about the relevance of the observed differences in drug interactions. The authors should select their favourite method (it indeed need not be flow cytometry) to measure growth (rather than survival via CFU) and show that what they measure in separation holds in co-culture. This should be done for at least a few drug pairs, including the most striking examples listed in Fig. 3. Without such experiments, their conclusion that the experimental outcome in a system where the two species are competing for the same nutrients is predictable from the experiments done in separation is not justified.

We thank the reviewer for their consideration suggesting a more appropriate validation method for our model. In our revision, we conducted flow cytometry experiments for co-cultures of two yeast species

for the most striking examples in Figure 3. In collaboration with the Khalil lab from Boston University, we showed that the Dyc+Cal and MMS+Rap combinations favor the growth of *C. albicans* over *S. cerevisiae* when compared to individual drug effects, in agreement with our theoretical framework. In the revised submission, we include these results as Figure 4 and Supplementary Figure 6. Our findings are described in the results as follows:

“Validation of the selectivity model with multiplexed drug-interaction testing

Here we have modeled the selectivity of combinations of drugs to different fungal species. However, it is worth noting that sensitivity to drug combinations was tested separately for each species, and not together. To test the predictive power of our model, we conducted co-culture drug interaction assays with fluorescently labeled strains of *S. cerevisiae* (mCherry) and *C. albicans* (GFP) treated with two individual drugs or their combination. We assessed the relative growth of each yeast species in the co-culture with flow cytometry (Figure 4a) for two drug pairs with striking phenotypes illustrated in Figure 3: (i) CAL+DYC is antagonistic against *C. albicans* and synergistic against *S. cerevisiae*. (ii) MMS+RAP is antagonistic in both species but the antagonism is stronger in *C. albicans*. Both CAL+DYC and MMS+RAP combinations selected for *C. albicans* growth compared to single drug treatment (Figure 4b and 4c). For each drug pair, we compared the observed percent of GFP labeled cells under combination treatment with the mean percent of GFP labeled cells under individual drug treatments ($n = 3$). This ratio was significantly larger than 1 for both CAL+DYC and MMS-Rap (t-test, p -values < 0.05), indicating that these drug combinations favor the growth of *C. albicans*, as predicted by our model (Supplementary Figure 6).”

We reported the details of these experiments in the methods section as follows:

“Multiplexed drug-interaction assay assessed by flow cytometry

S. cerevisiae (mCherry) and *C. albicans* (GFP) were grown in YPD liquid culture overnight at 30°C to $OD_{600} = 0.5$, diluted to $OD_{600} = 0.1$ and combined in equal volume. Cells were then co-incubated on 96-well plates in single drugs or 1:1 ratio combination of drugs. Each well had a final volume of 160 μ l with a solvent concentration of 2% DMSO. Cells were incubated for 4

hours shaking at 900 rpm, at 30°C. Cell/drug mixtures were then assessed for the relative abundance of each yeast species by flow cytometry. For all experimental conditions, >20,000 events were acquired using an Attune NxT Flow cytometer. Events were gated by forward and side scatter, and fluorescence distributions were calculated in FlowJo. Single cell cultures were used to define the gates for GFP-positive and mCherry-positive yeasts, representing *C. albicans* and *S. cerevisiae*, respectively.”

Reviewer #2 (Remarks to the Author):

The revised manuscript is much improved for clarity. The authors have thoroughly addressed all my questions from the initial review. However, I think the authors did not place properly their work in the context of prior studies (Lehar et al. 2009, Bulusu et al., 2009, Baym et al. 2016).

According to the authors, their main findings are the following:

1) Providing a model framework that can be broadly applied to other biomedical contexts wherein drug combinations may influence the therapeutic window or model the effects of multiple phenotypes of interest.

b) their model is broadly applicable because the required data is cell growth or death, which can be measured just as readily in human cells as in yeast.

a) their analytical model is in no way limited to specific microbes that can be cultured together.

It still remains unclear for me whether their work can really offer a considerably novel model framework for measuring selectivity of drug combinations compared to the recent investigations (Lehar et al. 2009, Bulusu et al., 2009). Is their selectivity score is more accurate or more broadly applicable compared to the ones presented earlier? They claim that their model is more broadly applicable because the required data is cell growth or death, which can be measured just as readily in human cells as in yeast and their model is in no way limited to specific microbes that can be cultured together. Are the previous selectivity models not based on cell growth or dead and are they limited to microbes that can be cultured together? If their model is more broadly applicable than the previous models can they provide some

sort of evidence for it? It would be important to present for the readership of the Nature Communications what are the real advantages or disadvantages of their selectivity model compared to the ones that are available in the literature.

We thank the reviewer for their valuable comments. While the reviewer here noted Bulusu et al. 2009, we will refer to the more encompassing Bulusu 2016 review in Drug Discovery Today for our response in keeping with our manuscript references and previous reviewer comments.

Following the reviewer's comment, in the revised manuscript we compared our study to these papers in more detail and strived to put our study in their context. The Lehar et al. 2009 article previously developed a selectivity index, similar to the selectivity index used in our study. We did not mean to suggest that the index used here is more broadly applicable than that of Lehar et al. 2009. Rather, we had meant to note that selectivity index is more broadly applicable than what can be observed in a coculture experiment. We regret the previously vague wording, and have edited this sentence for clarification.

Lehar et al correctly identifies that for therapeutic selectivity, synergy against a target must not be accompanied with synergistic toxicity. However, this article does not mention antagonism as a possible source of therapeutic selectivity. Bulusu et al. 2016 discusses the clinical relevance of selectivity over simply drug synergy, with only one reference: Lehar 2009. Neither study mentions a possible relationship between antagonism and enhanced selectivity. Our manuscript extends the idea presented in these studies and demonstrates that differential interactions underlie changes in therapeutic selectivity. Baym 2016 is a review article; it hypothesizes that selectivity between sensitive and resistant strains of the same species might be achieved via difference of drug interactions. In our study, we present the first experimental evidence of this phenomenon by using a large-scale screen of drug interactions in two species, generating a framework to calculate selectivity change in drug combinations following the example of Lehar 2009, and prospective validation of the clinically relevant phenotypes of cell growth and death.

Therefore, our study generalizes the conclusions of Lehar 2009 and is in agreement with the conjecture in Baym 2016, concluding that therapeutic selectivity of a combination is not necessarily only due to

synergy or antagonism, but is based on the difference of drug interactions for intended and side-effects. In our revision, we clarified the context of our study within Lehar 2009, Bulusu 2016 and Baym 2016 articles.

2) The magnitude of difference in interactions (including antagonism) defines the selectivity of drug combinations is a novel contribution.

I do not think that this is a strong enough claim to be considered as a novelty. Baym et al., 2016 not just discusses how drug interactions can affect the evolution of drug resistance within a single species, but also discusses how can differential drug interactions select against the resistant strain. Baym et al. 2016 also clearly show the concentration regime in the two-drug concentration space that can select against the resistant strain (Baym et al. 2016 Science, Fig 2. Selection inversion approaches and potential strategies.)

While we respect and cite the hypothetical idea described in Figure 2 of the Baym 2016 review article—that differential drug interactions may select against drug-resistant strains—we note that this was presented as a conjecture without experimental support. Our manuscript provides the experimental evidence to support this prediction, demonstrating the widespread presence of differential drug interactions between different cell types that influence cell-type selectivity of drug treatments.

I think the most novel aspect of the paper is the demonstration of the multiplexed drug interaction assay. Their study provides the first demonstration of a multiplexed drug interaction assay, where the interaction is simultaneously determined for multiple species in a heterogeneous culture. But, they claim that while their co-culture assay is an exciting means to detect selectivity, their analytical model is more broadly applicable to clinically relevant selectivity than specific microbes that may be cultured simultaneously. However, I believe, that their approach would be very useful to measure drug interaction simultaneously for drug resistant and sensitive microbes in order to identify concentration regimes of the two-drug space that specifically select against drug-resistant strains. In this case, a mixed-culture experiment would be biologically relevant and co-culturing resistant and sensitive strains of the same species would not be a problem.

We thank the reviewer for these encouraging comments. In the revised manuscript, we extended our multiplexed drug interaction assay by using flow cytometry. We had initially measured two drug interactions simultaneously in two species for the phenotype of fungicidal activity. For the revision, we developed a new assay for simultaneous interaction assessment where we mixed fluorescent-labeled yeasts, treated with drugs and their combinations, and used flow cytometry to count the relative abundance of each species after four hours of co-culture. This assay provides a fast and efficient means to measure selectivity change in drug combinations. In addition, this multiplexed drug interaction assay is also applicable to fungistatic compounds. Using this assay, we validated selectivity inversion via differential drug interactions for two striking examples from Figure 3. In the revised manuscript, we include these results as Figure 4. We agree that the simultaneous measurement of interactions in sensitive versus resistant strains would be another useful application of our multiplex drug interaction assays and have included this idea in our discussion.

Reviewers' comments:

Reviewer #1 (Remarks to the Author):

Weinstein and colleagues have revised their manuscript which now includes a co-culture assay to validate two of their most striking predictions based on single-species drug interaction assays. This validation could strengthen the generalisations they are making. However, the results from the co-culture assay are not fully conclusive and there are several technical issues of the flow-cytometry experiment that need to be clarified:

1. There is a sizable GFP-negative and mCherry-negative population in the Dyc-Cal flow cytometry experiment (bottom left quadrant in Figure 4b). Are these dead *S. cerevisiae* (or maybe *C. albicans*) cells? It would be important to clarify what this population is and explain it in the text.
2. More importantly, the mCherry marginal distributions shown in Fig. 4 as well as Fig. S6 give the impression that this GFP-neg mCherry-neg population was included in the analysis. This would be problematic: the correct way to compare the populations of *S. cerevisiae* vs *C. albicans* would be to compare the mCherry-positive & GFP-negative (top left quadrant) vs. mCherry-negative & GFP-positive (bottom right quadrant) populations only. It is thus not clear that this experiment was analysed correctly; the correct analysis might change the conclusions of the experiment. If marginal distributions are shown in the plots, this should be done for both mCherry and GFP. If the analysis was done comparing mCherry-pos GFP-neg vs. mCherry-neg GFP-pos populations, this needs to be clearly indicated in the plots and explained in the methods or figure legends.
3. While the Dyc-Cal co-culture experiment is qualitatively in agreement with the expectation based on the data in Figure 3e, it should be possible to make a more quantitative prediction. E.g. in Supplementary fig. 6 the authors only plot the Q1 GFP+ fraction which indicates that the fraction of *C. albicans* in total cytometer events has increased. This supports the conclusion of the single-species experiments qualitatively, but can the authors make some quantitative prediction based on the delta-alpha determined in the single-species experiment and compare this to the flow cytometry data?
4. More importantly, the results of the MMS-Rap co-culture experiment shown in Figure 4c do not agree with the expectation from the single-species experiment (Figure 3c). According to the latter, *C. albicans* should have a clear growth advantage in all conditions, including the single drug conditions; however, the *S. cerevisiae* population looks larger in both MMS and Rap alone and this is confirmed in Suppl. Fig. 6b where the Q1 GFP+ fraction is consistently below 50% in these conditions.
5. The co-culture experiment was run for an extremely short time which may be insufficient for the full effect of the drugs to kick in. This assay ran for only 4h after mixing the two species; given that growth is partially inhibited by the drugs, this is hardly one doubling time (in contrast, the drug interaction assays were run for 16h which is more reasonable). It is standard procedure in microbiology to allow for at least ~7 generations in such assays so that the cells have sufficient time to adapt to the new conditions and can reach a new steady state of growth. E.g. similar experiments competing sensitive and resistant *E. coli* strains were run for 24h in (Chait et al., Nature, 2007), see Fig. 3 in that paper. In brief, to be convincing, this assay would need to be run for a similar time as the drug interaction assays.

Reviewers' comments:

Reviewer #1 (Remarks to the Author):

Weinstein and colleagues have revised their manuscript which now includes a co-culture assay to validate two of their most striking predictions based on single-species drug interaction assays. This validation could strengthen the generalisations they are making. However, the results from the co-culture assay are not fully conclusive and there are several technical issues of the flow-cytometry experiment that need to be clarified:

We thank the reviewer for these comments. In our revised submission, we provided additional technical details to clarify our flow cytometry experiments and conducted additional analysis. We believe that the revised manuscript convincingly conveys the message that the selectivity of drug combinations is influenced by the difference between drug interactions. Our co-culture experiments validate the predicted drug interaction dependent selectivity for two drug pairs, with synergistic or antagonistic interactions.

1. There is a sizable GFP-negative and mCherry-negative population in the Dyc-Cal flow cytometry experiment (bottom left quadrant in Figure 4b). Are these dead *S. cerevisiae* (or maybe *C. albicans*) cells? It would be important to clarify what this population is and explain it in the text.

Since the population in the GFP-/mCherry- quadrant carries none of the fluorescent markers, we assume that this population of cells consists of dead, dying or overall poorly protein producing cells.

2. More importantly, the mCherry marginal distributions shown in Fig. 4 as well as Fig. S6 give the impression that this GFP-neg mCherry-neg population was included in the analysis. This would be problematic: the correct way to compare the populations of *S. cerevisiae* vs *C. albicans* would be to compare the mCherry-positive & GFP-negative (top left quadrant) vs. mCherry-negative & GFP-positive (bottom right quadrant) populations only. It is thus not clear that this experiment was analysed correctly; the correct analysis might change the conclusions of the experiment. If marginal distributions are shown in the plots, this should be done for both mCherry and GFP. If the analysis was done comparing mCherry-pos GFP-neg vs. mCherry-neg GFP-pos populations, this needs to be clearly indicated in the plots and explained in the methods or figure legends.

Thank you for these astute observations. We apologize for not including this detail in our previous submission. We ignored GFP-/mCherry- cells while computing delta-%*C. albicans* scores. Therefore, we have already used GFP+/mCherry- cells as *C. albicans* and mCherry+/GFP- cells as *S. cerevisiae* populations, in line with the reviewer's comments on the correct populations for comparison. We corrected the labels in Supplementary Figure 6 from %GFP+ to %*C. albicans*.

We agree with the reviewer that the cells in bottom left and top right quadrants confound the interpretation of the histograms given in Figure 4 and Supplementary Figure S6. In the revised submission, we transparently masked these quadrants in the figures and removed these cells from the histograms. In our initial submission, we had shown only the marginal distributions for mCherry signal for visual clarity. In the revised submission, we also show the GFP marginal distributions for each experiment in Supplementary Figure 6.

3. While the Dyc-Cal co-culture experiment is qualitatively in agreement with the expectation based on the data in Figure 3e, it should be possible to make a more quantitative prediction. E.g. in Supplementary fig. 6 the authors only plot the Q1 GFP+ fraction which indicates that the fraction of *C. albicans* in total cytometer events has increased. This supports the conclusion of the single-species experiments qualitatively, but can the authors make some quantitative prediction based on the delta-alpha determined in the single-species experiment and compare this to the flow cytometry data?

We thank the reviewer for this excellent suggestion. In our previous submission, we had generated delta-alpha scores for each drug combination by using $\alpha_{alb} - \alpha_{cer}$ using the single-species experiments. However, we haven't explicitly compared delta-alpha and flow cytometry data. In the revised submission, we note that delta-alpha scores are quantitative predictions regarding the selectivity due to drug interactions. The delta-alpha scores of Dyc-Cal and MMS-Rap are 1.1 and 0.7, respectively, suggesting that these combinations will favor the growth of *C. albicans* cells in comparison with single drug selectivity.

In our previous submission, we had also included a quantitative assessment of the selectivity in co-culture experiments, by assessing the average of %*C. albicans* in two single drugs to define an expected %*C. albicans* in the combination treatment. We used the observed %*C. albicans* minus the expected %*C. albicans* (delta-%*C. albicans*) as a measure for selectivity change due to drug interactions. This score is 0 if the combination has no selectivity due to drug interactions, positive or negative if there is selectivity favoring *C. albicans* or *S. cerevisiae*, respectively. delta-%*C. albicans* scores were significantly larger than 0 in both experiments (16 and 9 for Dyc-Cal and MMS-Rap, respectively), supporting the model predictions that the drug combinations

favor the growth of *C. albicans* cells compared to single drugs. Following the reviewer's comment, in the revised submission we explicitly compared the delta-alpha scores for Dyc-Cal and MMS-Rap with delta-%*C. albicans* scores obtained from co-culture experiments.

4. More importantly, the results of the MMS-Rap co-culture experiment shown in Figure 4c do not agree with the expectation from the single-species experiment (Figure 3c). According to the latter, *C. albicans* should have a clear growth advantage in all conditions, including the single drug conditions; however, the *S. cerevisiae* population looks larger in both MMS and Rap alone and this is confirmed in Suppl. Fig. 6b where the Q1 GFP+ fraction is consistently below 50% in these conditions.

We thank the reviewer for providing an opportunity to clarify our methodology. Our co-culture experiments are designed to measure a *change* in selectivity due to drug interactions, and do not provide information on single drug or combination selectivity. The individual drug selectivity in the co-culture experiments may vary from the single species experiments due to experimental variation. While the mixed yeast cultures have equal optical densities, they may not contain equal number of colony forming units. Therefore, the assumption that %*C. albicans* has dropped from 50% may not always be true and would require a different experimental setup to measure. However, the comparison of the %*C. albicans* in combination with an expected %*C. albicans* given single drug effects allows us to verify the model prediction that selectivity may change due to difference of drug interactions. Cal-Dyc and MMS-Rap combinations favor the growth of *C. albicans* as compared to single drug selectivity, although the former is synergistic in both species and the latter is antagonistic in both species. The delta-alpha scores of both these pairs are high, suggesting *C. albicans* selectivity, in agreement with the co-culture delta-%*C. albicans* scores.

5. The co-culture experiment was run for an extremely short time which may be insufficient for the full effect of the drugs to kick in. This assay ran for only 4h after mixing the two species; given that growth is partially inhibited by the drugs, this is hardly one doubling time (in contrast, the drug interaction assays were run for 16h which is more reasonable). It is standard procedure in microbiology to allow for at least ~7 generations in such assays so that the cells have sufficient time to adapt to the new conditions and can reach a new steady state of growth. E.g. similar experiments competing sensitive and resistant *E. coli* strains were run for 24h in (Chait et al., Nature, 2007), see Fig. 3 in that paper. In brief, to be convincing, this assay would need to be run for a similar time as the drug interaction assays.

While it is not within the scope of our analysis to determine if the full effect of the drugs take place, we importantly note that the observed effects from this short exposure agree with the model predictions. In addition, our flow cytometry experiments clearly demonstrate that the distribution of cells significantly change in single drug or combination treatments in this short time. We opted not to grow our co-cultures for a longer period, due to the difference of doubling times between the yeast species, with *C. albicans* doubling every 2 hours, whereas *S. cerevisiae* doubles every 2.5 hours. This contrasts with the co-culture experiments shown in Chait et al., which uses mutant versions of same species with similar fitness. We used cells that are in log-phase and ended our experiments at ~ 2 doubling times, aiming to observe selectivity due to drug interactions rather than proliferation rate. Following the reviewer's comments, we computed the drug interaction scores for all *C. albicans* and *S. cerevisiae* experiments by using only a 4-hour subset of the growth data from the log-phase, which more closely correspond to our validation experiments. These scores significantly correlated with the scores obtained using the full growth data, further supporting the relevance of the use of a 4-hour co-culture experiment. In addition, when only 4 hours of the growth data was used, the delta-alpha scores for Cal-Dyc and MMS-Rap were positive, in agreement with the co-culture delta-%*C. albicans* scores.

Reviewers' comments:

Reviewer #1 (Remarks to the Author):

The revised manuscript by Weinstein and colleagues clarifies the points raised in the previous round of review with two exceptions:

1. In Supplementary Fig. 6, the authors compare the prediction from a single-drug condition to the outcome in the co-culture flow cytometry experiment. However, my previous point #3 asked for quantitative comparison of the prediction from the single-species experiments (that is, single-species experiment in drug combination as in Fig. 3) to the outcome of the flow-cytometry co-culture experiment. The purpose of this was to seek validation of the quantitative measurements made by the authors, rather than to demonstrate that drugs interact, leading to a large difference between a prediction from single-drug condition and the measurement in the drug combination. This was probably a misunderstanding but the issue remains unresolved.

2. I do not understand the authors' reply to my previous point #4: Fig. 3c shows that MMS alone, RAP alone, and MMS-RAP should all strongly select for *C. albicans*. The co-culture experiment in Fig. 4c shows weak selection for *C. albicans* in the drug combination but the single drug conditions seem to show similarly weak selection for *S. cerevisiae*. Why should we consider the former a major result and the latter a consequence of "experimental variation" or of different initial cell numbers for *C. albicans* and *S. cerevisiae*? What is the "different experimental setup" that would be required to measure this properly? It seems straightforward to do a properly controlled experiment for this purpose using the existing setup. I understand that the quantitative analysis shows some enrichment of *C. albicans* compared to the single-drug effects but the relevance of such subtle differences also remains unclear, given that each of the drugs individually can be used to select for *C. albicans*. This point is clearer for DYC-CAL where selection for *C. albicans* seems to work only in the combination of both drugs.

Reviewers' comments:

Reviewer #1 (Remarks to the Author):

The revised manuscript by Weinstein and colleagues clarifies the points raised in the previous round of review with two exceptions:

1. In Supplementary Fig. 6, the authors compare the prediction from a single-drug condition to the outcome in the co-culture flow cytometry experiment. However, my previous point #3 asked for quantitative comparison of the prediction from the single-species experiments (that is, single-species experiment in drug combination as in Fig. 3) to the outcome of the flow-cytometry co-culture experiment. The purpose of this was to seek validation of the quantitative measurements made by the authors, rather than to demonstrate that drugs interact, leading to a large difference between a prediction from single-drug condition and the measurement in the drug combination. This was probably a misunderstanding but the issue remains unresolved.

We thank the reviewer for additional clarification for their request. In the revised submission, we conducted two additional controls in our flow cytometry experiment to allow a quantitative comparison from single species experiments and mixed culture experiments. First, we mixed the cells based on cell counts rather than OD, with approximately equal number of cells from both species based on flow cytometry. Second, we included a no drug condition for the mixed cultures; by this, we could verify the single drug selectivity for each drug. Comparison of the % C.albicans in the no drug experiment with the %C.albicans in t0 shows that %C.albicans increase with time, which is expected since C.albicans has a faster growth rate than S.cerevisiae. Comparison of the selectivity in single drugs or the combination with the

selectivity in no drug combination indicates that (i) CAL and DYC select for *S.cerevisiae*, however CAL+DYC selects for *C.albicans*; and (ii) MMS and RAP both select for *C. albicans*, and MMS+RAP selects for *C.albicans* with greater strength. These results are presented in summary form in Figure 4 and in detail in a revised Supplementary Figure 6. We believe that the use of a no drug condition in our flow cytometry experiments resolved this concern.

2. I do not understand the authors' reply to my previous point #4: Fig. 3c shows that MMS alone, RAP alone, and MMS-RAP should all strongly select for *C. albicans*. The co-culture experiment in Fig. 4c shows weak selection for *C. albicans* in the drug combination but the single drug conditions seem to show similarly weak selection for *S. cerevisiae*. Why should we consider the former a major result and the latter a consequence of "experimental variation" or of different initial cell numbers for *C. albicans* and *S. cerevisiae*? What is the "different experimental setup" that would be required to measure this properly? It seems straightforward to do a properly controlled experiment for this purpose using the existing setup. I understand that the quantitative analysis shows some enrichment of *C. albicans* compared to the single-drug effects but the relevance of such subtle differences also remains unclear, given that each of the drugs individually can be used to select for *C. albicans*. This point is clearer for DYC-CAL where selection for *C. albicans* seems to work only in the combination of both drugs.

We apologize for failing to be clear about this point: In our previous submission, we have not made any claims about single drug selectivity in the co-culture experiment. Our previous submission only aimed to compare the selectivity in the combination with the selectivity in single drugs, and this comparison agreed with model predictions. The reviewer's suggestion that for MMS and RAP "single drug conditions seem to show similarly weak selection for *S. cerevisiae*" stems from a misunderstanding that the coculture had equal number of cells from each species. However, as we explained in our responses, we had previously used equal OD of each species in mixed culture, rather than cell number. Following the reviewer comments we used the different experimental setup to address the reviewer's concern. We used flow cytometry data at the start of the experiment to verify that the cell mixture had two yeasts in approximately equal numbers. We also included a no drug condition to account for differential growth rates between yeasts. Using this setup, we could compare the single drug selectivity with the no drug condition to verify model predictions for single drug selectivity. In our revised submission, we followed the reviewer's advice and performed this experiment. This new experimental setup indicated that both the single drug and combination selectivity agree with our model predictions.